# Mapping the safe operating space of marine ecosystems under contrasting emission pathways

Timothée Bourgeois[1], Giang T. Tran[2], Aurich Jeltsch-Thömmes[3,4], Jörg Schwinger[1], Friederike Fröb[5], Thomas L. Frölicher[3,4], Thorsten Blenckner[6], Olivier Torres[7], Jean Negrel[1], David P. Keller[2,8], Andreas Oschlies[2], Laurent Bopp[7], Fortunat Joos[3,4]

[1]NORCE Research, Bjerknes Centre for Climate Research, Bergen, 5005, Norway
[2]Marine Biogeochemical Modelling, GEOMAR Helmholtz-Zentrum für Ozeanforschung Kiel, Kiel, 24105, Germany
[3]Climate and Environmental Physics, Physics Institute, University of Bern, Bern, 3012, Switzerland
[4]Oeschger Centre for Climate Change Research, University of Bern, Bern, 3012, Switzerland
[5]Geophysical Institute, University of Bergen, Bjerknes Centre for Climate Research, Bergen, 5005, Norway
[6]Stockholm Resilience Centre, Stockholm University, Stockholm, 106 91, Sweden
[7]LMD-IPSL, CNRS, Ecole Normale Supérieure/PSL Res. Univ, Ecole Polytechnique, Sorbonne Université, Paris, 75005, France
[8]Carbon to Sea Initiative, Washington, DC, USA

*Correspondence to*: Timothée Bourgeois (tbou@norceresearch.no)

**Abstract.** Anthropogenic greenhouse gas emissions cause multiple changes in the ocean and its ecosystems through climate change and ocean acidification. These changes can occur progressively with rising atmospheric carbon dioxide concentrations, but there is also the possibility of large-scale abrupt, and/or potentially irreversible changes, which would leave limited opportunity for marine ecosystems to adapt. Such changes, either progressive or abrupt, pose a threat to biodiversity, food security, and human societies. However, it remains notoriously difficult to determine exact limits of a "safe operating space" for humanity. Here, we map, for a variety of ocean impact metrics, the crossing of limits, which we define using the available literature and to represent a wide range of deviations from the unperturbed state. We assess the crossing of these limits in three future emission pathways: two climate mitigation scenarios, including an overshoot scenario, and one high-emission no-mitigation scenario. These scenarios are simulated by the latest generation of Earth system models and large perturbed-parameter ensembles with two Earth system models of intermediate complexity. Using this comprehensive model database, we estimate the timing and warming level at which 15 different impact metrics exceed 4 limits, along with an assessment of the associated uncertainties. We find that under the high-emissions scenario, the strongest severity of impacts is expected with high probability for marine heatwaves' duration, loss of Arctic summer sea ice extent, expansion of ocean areas that are undersaturated with respect to aragonite, and decrease in plankton biomass. The probability of exceeding a given limit generally decreases clearly under low-emissions scenario. Yet, exceedance of ambitious limits related to steric sea level rise, Arctic summer sea ice extent, Arctic aragonite undersaturation, and plankton biomass are projected to be difficult to avoid (high probability) even under the low-emissions scenario. Compared to the high-emissions scenario, the scenario including a temporary overshoot reduces with high probability the risk of exceeding limits by year 2100 related to marine heatwave duration, Arctic summer sea ice extent, strength of the Atlantic meridional overturning circulation, aragonite undersaturation,

global deoxygenation, plankton biomass, and metabolic index. Our study highlights the urgent need for ambitious mitigation efforts to drastically minimize extensive impacts and potentially irreversible changes to the world's ocean ecosystems.

## 1 Introduction

Earth system models (ESMs) are invaluable tools to simulate the climate outcomes of future emission pathways. However, due to computational constraints, ESMs are limited in terms of spatial resolution and complexity of represented processes (Chen et al., 2021). Consequently, it remains notoriously difficult to assess climate change impacts that occur at smaller scales or in systems that are not exhaustively represented in ESMs, especially for essential variables linked to marine biodiversity and marine ecosystems services (Pereira et al., 2013; Balvanera et al., 2022). Large-scale changes in important drivers of marine ecosystem processes (for example, warming, deoxygenation, and acidification) are often taken as a measure of potential ecosystem damage ("ecosystem stressors"). As such changes will usually occur simultaneously, the term "multiple potential ecosystem stressors" has been coined to describe the threat that climate change poses to marine ecosystems (Bopp et al., 2013; Gattuso et al., 2015; Gruber, 2011; Kwiatkowski et al., 2020). To consider the various stressors and management strategies that affect the Earth system, the concept of safe operating space has been proposed (Rockström et al., 2009). For the ocean, a safe operating space refers to the conditions under which marine ecosystems can remain resilient and continue to provide essential services despite ongoing environmental changes and human activities (Nash et al., 2017).

Although it is generally well-known which climate variables will have an adverse impact on ecosystems and societies if they are altered by human activity, exact limits that should not be exceeded are often difficult to define. This might be the case either because impacts occur gradually in synchrony with changes in a driver variable (and it thus remains an ethical or economic question how much damage can be accepted), or because a limit exists but it is highly uncertain. The latter will be the case if tipping points exist in the system, a crossing of which will lead to large and irreversible changes (e.g., Lenton et al., 2008; Armstrong McKay et al., 2022). For ecosystems in particular, the possibility exists that gradual changes in the physical or biogeochemical state may lead to the crossing of tipping points (Heinze et al., 2021). Nevertheless, our knowledge on the impacts of these changes on marine ecosystems is growing. Thermal changes induced by global warming are altering the productivity of some phytoplankton functional types and are reshaping established interspecific competition in marine ecosystems (Kordas et al., 2011; Dutkiewicz et al., 2013; Anderson et al., 2021). The shoaling of the calcium carbonate ($CaCO_3$) saturation horizon due to ocean acidification is threatening calcifying organisms (Orr et al., 2005; Doney et al., 2020). Ocean deoxygenation and the expansion of oxygen minimum zones contribute to marine aerobic habitat loss (Diaz and Rosenberg, 2008; Pinsky et al., 2020; Morée et al., 2023; Fröb et al., 2024). Shifting circulation patterns affect fish migrations and human societies (Van Gennip et al., 2017; Schwinger et al., 2022). Finally, upper-ocean stratification changes alter ocean primary productivity and community structures by exacerbating surface nutrient depletion (e.g., Fu et al., 2016).

In this study, we define a set of 15 impact metrics associated with 4 limits following the approach of Steinacher et al. (2013) to answer questions of the type:

- Under a high-emissions scenario, how likely is it for global mean steric sea level to rise by 0.4 m compared to 1850–1900?
- When and at which warming level will that happen?

To answer this type of questions we (1) define a set of large-scale metrics that indicate a threat for ocean ecosystems and/or human systems due to climate change, (2) attribute limits to each metric, from ambitious (challenging to stay within) to more relaxed, which translates into an increase in expected severity of impacts, and (3) explore projections of these metrics in scenario simulations. We use scenario simulations from two types of models: (1) nine state-of-the-art Earth system models from the latest Coupled Model Intercomparison Project (CMIP6), and (2) two perturbed parameter ensembles from Earth

system models of intermediate complexity (EMICs). EMICs are an important modelling tool in climate sciences due to their relatively low computational cost compared to ESMs. This advantage makes them suitable for generating large ensembles to quantify uncertainty (Steinacher et al., 2013; Steinacher and Joos, 2016), and for conducting long simulations, spanning several thousand years (e.g., Battaglia and Joos, 2018; Plattner et al., 2008). EMICs have also been extensively used to investigate the Earth system response to strong mitigation scenarios and to carbon dioxide removal (e.g. Jeltsch-Thömmes et al., 2024;

Tokarska et al., 2019). Both model types (ESMs and EMICs) have their specific advantages and limitations. The low computational demand of EMICs comes at the cost of resolution and complexity, which remains relatively low. ESMs with their higher resolution and complexity, can usually only be run for a limited number of ensemble members, if any. The CMIP6 ensemble of ESMs is an ensemble of opportunity, and the quantification of uncertainty using this ensemble has, although common practice, certain limitations (e.g., Knutti, 2010). Despite their importance, both model classes (EMICs and ESM) are

rarely compared. By applying the same analysis of impact metrics and associated limits to the same scenarios simulated by both model classes, this study aims to fill this gap.

## 2 Methods

### 2.1 Definition of impact metrics and limits

We use 15 illustrative impact metrics providing a broad spectrum of impacts of climate change on marine ecosystems, and

each of them are associated with 4 limits (Table 1). Limits are ranked according to the expected severity of impacts upon exceeding them: exceeding limit 4 for a given metric is expected to result in more severe impacts than exceeding limit 1. Thus, staying below limit 1 is more ambitious because a higher emission reduction would be required to achieve this goal. Limits at a given level are not necessarily dependent, i.e., they can be exceeded at different time and global warming levels.

A literature review combining observations with simulation studies on critical limits in the ocean system is conducted to define

the impact metrics and limits. While the aim of this study is to define limits based as much as possible on the literature, many metrics suffer from a lack of knowledge regarding the assessment of actual impacts that an exceedance would have on the Earth system or ecosystem functioning, especially in a multi-stressor context (Williamson and Guinder, 2021). Some physical metrics have been more thoroughly investigated, while biogeochemical metrics are less constrained. Observations and

laboratory experiments suggest numerous critical limits for key ecosystem stressors. Moreover, these limits are species-
dependent and can vary over a wide range. Thus, for some metrics, we favour limits based on relative changes to characterise
reasonably safe levels instead of absolute changes. These choices could be refined through future research and further dialogue
with stakeholders. If the literature does not permit us to define limits for a specific metric, then we use *ad hoc* limits that cover
the simulated mitigation space from very strong mitigation to very little or no mitigation effort.

The impact metrics include six physical parameters, related to surface atmospheric warming, marine heatwaves, steric sea
level rise, sea ice extent, and the Atlantic Meridional Overturning Circulation (AMOC), five chemical parameters, related to
global and regional ocean acidification and deoxygenation, and four ecosystem parameters, related to productivity, biomass,
organic matter export, and metabolic performance.

**Physical metrics**

Global mean surface air temperature (SAT) is an important metric of the climate system and has strong and direct influences
on ecosystems as well as human systems, i.e., many other important indicators and metrics co-vary with temperature. We pick
the limits of 1.5°C and 2°C increase since the 1850–1900 mean based on the Paris agreement (UNFCCC, 2015). Two additional
limits of 3°C and 4°C represent temperature limits beyond which severe impacts and the triggering of global tipping elements
could be possible (Armstrong McKay et al., 2022; Masson-Delmotte et al., 2021). These limits for global mean SAT increase
have also been previously used in Steinacher et al. (2013).

We consider marine heatwaves due to their substantial global and regional impacts on marine ecosystems (Capotondi et al.,
2024; Frölicher and Laufkötter, 2018; Smith et al., 2021). We use two definitions of marine heatwaves based on different
baselines (Burger et al., 2022; Smith et al., 2025). First, we define a marine heatwave day as the local daily mean sea surface
temperature (CMIP6 variable *tos*) exceeding the 90[th] percentile relative to a fixed seasonally varying 1850–1900 baseline
(metric abbreviated $MHW_{fix}$). In this case, changes in marine heatwaves are driven by both long-term surface ocean warming
trends and changes in anthropogenically-forced internal variability. The fixed baseline may be particularly relevant for
assessing the risk marine heatwaves pose to organisms with slow adaptation rates. Second, we define a marine heatwave day
relative to a shifting-mean baseline ($MHW_{shift}$), where the 1850–1900 percentile thresholds are adjusted according to the forced
mean trend in sea surface temperature (SST). The forced trend is identified using a smoothing "Enting" spline (Enting, 1987)
with a 80-yr cut off period. In the shifting-mean approach, changes in marine heatwave duration are primarily driven by
changes in anthropogenically-forced internal variability, while the long-term warming trends is already accounted for in the
baseline (Burger et al., 2022; Deser et al., 2024). The choice between a fixed or shifting baseline depends on the specific
application. For example, the shifting-mean case may better capture the risks posed to organisms that can adapt to long-term
warming trends. For both definitions, we (1) calculate the global annual mean duration of marine heatwaves, and (2) deduce
the anomaly relative to the 1850–1900 period to normalize model-dependent internal variability. Given the lack of
observational constraints on global marine heatwaves, we distribute the limit values of $MHW_{fix}$ uniformly over the year as 90,

180, 270, 360 days (the latter representing an almost permanent heatwave). We distribute the limits of MHW$_{shift}$ over the range of projected values under the scenarios used in this study with 4, 6, 8, and 10 days.

The third physical metric chosen is the rise of steric sea level (SSL). Sea level is rising at accelerating rates, which poses a significant challenge to coastal ecosystems and community livelihoods. We are only considering the SSL rise because the models used in our study do not simulate the melting of glaciers and ice sheets. While strongly connected to global warming, SSL rise shows a delayed response due to the long thermal lag of the ocean system (Levermann et al., 2013). The SSL rise is estimated to account for 40 % of the total sea-level rise of 0.2 m today, i.e., the current estimated SSL rise is estimated at around 0.08 m (Church et al., 2011; Fox-Kemper et al., 2021; WCRP Global Sea Level Budget Group, 2018). O'Neill et al. (2017) estimate that risks related to SSL rise are at a moderate level at about 0.1 m above the 1986-2005 level, and transition to high risks are expected at around 1 m above the same reference level. Hinkel et al. (2014) find that under no adaptation, 0.25–1.23 m of global sea-level rise in 2100 (i.e., 0.1–0.5 m of SSL rise assuming a constant steric fraction) would expose 0.2–4.6 % of the global population to flooding annually. Hermans et al. (2021) found a mean SSL rise of 0.27 m in 2100 simulated by CMIP5 and CMIP6 ensembles under high-emissions scenarios. According to Hague et al. (2023), the current sea level rise is already expected to increase flood frequencies at a level of 0.2 m of sea level rise (or 0.08 m of SSL rise). Thus, we chose to define the four limits as 0.1, 0.2, 0.3, and 0.4 m of SSL rise relative to the period 1850–1900, to encompass the range found in the above-mentioned literature.

Changes in summer Arctic sea-ice extent have a direct impact on the climate system through the albedo feedback. Furthermore, a substantial reduction of Arctic sea ice could threaten the livelihood of organisms that depend upon habitats provided by sea ice. Arctic sea ice is projected to decline, and an ice-free summer state is expected even with a stabilised global warming of 1.5°C (~1 % chance of individual ice-free years by the end of the century; Pörtner et al., 2019). Here, the summer ice-free state is defined as a September sea ice extent below $10^6$ km$^2$. We further define three more limits up to 4 x $10^6$ km$^2$ following the projected range from Stroeve et al. (2012) and Peng et al. (2020). Exceedances of the 4 x $10^6$ km$^2$ limit has already been observed in 2012 and 2020 (https://nsidc.org/, visited on May 21[st], 2025).

A collapse of the Atlantic meridional overturning circulation (AMOC) is often considered a more distant tipping point (Lenton, 2012) even though recent literature estimated that we cannot rule out that AMOC is on course to collapse (Van Westen et al., 2024). The estimated probabilities from expert elicitation for a shutdown of AMOC (until 2100) is 0–0.2 for low (<2°C), 0–0.6 for medium (2–4°C), and 0.05–0.95 for high climate change (4–8°C), according to Zickfeld et al. (2007) and Kriegler et al. (2009). A weakening of the AMOC this century is expected (Pörtner et al., 2019; Fox-Kemper et al., 2021), which can cause, for example, changes in the rainfall, storm frequency in Northern Europe and a decrease in marine productivity in the North Atlantic (Pörtner et al., 2019). We compute the strength of the AMOC as the vertical maximum of the stream function at 26°N following Weijer et al. (2020). As for the limits, despite a growing body of literature on the historical and projected evolution of the AMOC, we still lack sufficiently long observation-based time series, knowledge, and scientific consensus to understand if the AMOC is already experiencing a decline exceeding natural variability, and if such a decline is attributed to anthropogenic forcing (Jackson et al., 2022; Latif et al., 2022; Lobelle et al., 2020; Terhaar et al., 2025). Due to the absence

of more robust knowledge, we choose four limits at 20, 25, 30, and 40 % decline relative to 1850–1900 to cover the range of model responses (Weaver et al., 2012; Weijer et al., 2020).

**Ocean acidification metrics**

Ocean omega aragonite ($\Omega_{arag}$), or the level of saturation of the least-stable form of calcium carbonate in seawater, is a common indicator of the potential for biotic calcification (Gazeau et al., 2007). Ocean acidification could lead to undersaturation ($\Omega_{arag}<1$) and dissolution of calcium carbonate in parts of the surface ocean during the 21[st] century, which can have detrimental effects on marine ecosystems (Orr et al., 2005). Studies have shown that no prominent present-day coral reefs exist in environments with $\Omega_{arag}<3$ (Guinotte et al., 2003; Hoegh-Guldberg et al., 2007; Kleypas et al., 1999a). A lower limit of

$\Omega_{arag}<1.5$ has been used previously to indicate water masses which may be stressful to larvae of shellfish such as oysters (Ekstrom et al., 2015; Gimenez et al., 2018). For $\Omega_{arag}<1.5$, calcifying organisms have trouble forming shells during the first few days of their life (Waldbusser et al., 2015). Guinotte et al. (2006) estimated that over 95 % of cold water biotherm-forming corals were found in water masses that were supersaturated ($\Omega_{arag}>1$). We define three ocean acidification metrics in terms of area fractions. The first two metrics, abbreviated $A_{SO}$ and $A_{Arctic}$, are respectively the surface area fractions of the Southern

Ocean (Nouth of 50°S) and the Arctic Ocean (North of 70°N) undersaturated with respect to aragonite ($\Omega_{arag}<1$; annual mean), which means that seawater becomes corrosive to aragonitic shells (Doney et al., 2009; Fabry et al., 2009). The selected limits for these metric range from 20 % to 80 % following Steinacher et al. (2009, 2013). The third ocean acidification metric, $A_{\Omega>3}$, addresses areas with high saturation states ($\Omega_{arag}>3$) that are mainly found in the tropics and subtropics (Kleypas et al., 1999b). We define this variable as the percentage of the global ocean surface area with $\Omega_{arag}>3$ that has been lost since pre-industrial

times and select limits from 50 % to 100 % (Steinacher et al., 2013). For the sake of readability, these metrics are sometimes referred in the text to Southern Ocean $\Omega_a<1$, Arctic Ocean $\Omega_a<1$ and Global Ocean $\Omega_a>3$, respectively.

**Other biogeochemical metrics**

Marine species have been observed to die after exposure to a wide range of critical oxygen ($O_2$) levels, from 8.6 mg $O_2$ L$^{-1}$ (ca.

275 µmol L$^{-1}$) to anoxia (Vaquer-Sunyer and Duarte, 2008). Critical $O_2$ levels are largely species- and stage-specific (Ekau et al., 2010), making it challenging to define common limits. Globally, dissolved $O_2$ is projected to decline by 1.81 to 3.45 % by the year 2100 under CMIP5 representative concentration pathways (RCP) (Bopp et al., 2013). Subsurface (100-600 m) $O_2$ is projected to decline by 3.1-4 % under RCP8.5 and 0.1-0.5 % under RCP2.6. The projected decline in the subsurface dissolved $O_2$ concentration for CMIP6 models under their shared socioeconomic pathways (SSP) vary from -6.36 to -13.27 µmol L$^{-1}$ by

the end of the century (Kwiatkowski et al., 2020). The equivalent range for CMIP5 models under RCP scenarios is from -3.71 to -9.51 µmol L$^{-1}$. Due to large differences between models and when compared to observations, we decided to define relative limits for two metrics: mean global full-depth $O_2$ concentration and volume of hypoxic waters above 1000 m depth (i.e., waters with <63 µmol L$^{-1}$; Limburg et al., 2020).

The decline of marine net primary productivity (NPP) is considered one of the primary stressors of open ocean ecosystems (Bopp et al., 2013). Limits for marine NPP are defined in terms of relative changes for the same reason as above. Kwiatkowski et al. (2020) shows changes from -0.56 ± 4.12 % under SSP1-2.6 to -2.99 % ± 9.11 % under SSP5-8.5 for CMIP6 models, while the range for CMIP5 models is from -3.42 ± 2.47 % under RCP2.6 to -8.54 ± 5.88 % under RCP8.5 until year 2100. Thus, we set the limits to 2, 3.5, 4 and 8 % relative to 1850–1900. In addition to NPP, we consider changes in plankton biomass (ΔBiomass) because projected plankton biomass has been considered as a more robust metric reflecting the impact of climate change on marine ecosystems (Bopp et al., 2022; Tittensor et al., 2021). The ΔBiomass metric represents the change in the sum of phytoplankton and zooplankton biomass (CMIP6 variables *zooc* and *phyc*) and its limits are the same as those for NPP. We excluded the model CNRM-ESM2-1 for this metric due to a large inconsistent variability found over the historical period, which has been attributed to mesozooplankton biomass.

We consider changes in the upper ocean metabolic index and changes in particulate organic matter (POM) export between 30°S and 30°N as indicators of the compound effects of warming and oxygen changes on viable habitat and the survival of marine species (Battaglia and Joos, 2018). The metabolic index, Φ, is defined as the ratio of $O_2$ supply to an organism's resting $O_2$ demand. Warming ocean and lower partial pressure of $O_2$ is expected to reduce the globally averaged upper ocean (0–200 m) metabolic index, which was shown to restrict viable habitats (Deutsch et al., 2015). Φ has been calculated following Fröb et al. (2024) using the median ecophysiotype of the 61 species described in Deutsch et al. (2020), without considering biomass distribution. The export of POM is the primary food source for deep-sea organisms. Thus, the POM export between 30°S and 30°N serves as an indicator of food availability in deep sea habitats. The limits of 4, 6, 8, and 10 % for these two indicators are based on the result of Battaglia and Joos (2018).

**Table 1. Impact metrics and corresponding limits for changes until year 2100. Limits that are considered for ESMs only are marked with an asterisk.**

| Impact metric | Description | Level 1 | Level 2 | Level 3 | Level 4 | Unit |
|---|---|---|---|---|---|---|
| ΔSAT | Increase in mean annual global surface atmospheric temperature relative to 1850–1900 | 1.5 | 2 | 3 | 4 | °C |
| MHW$_{fix}$* | Global mean duration of marine heatwaves within a year, relative to a fixed baseline | 90 | 180 | 270 | 360 | day |
| MHW$_{shift}$* | Same as MHW$_{fix}$ (line above), but using a shifting baseline approach | 4 | 6 | 8 | 10 | day |
| ΔSSL | Mean annual steric sea level rise relative to 1850–1900 | 0.1 | 0.2 | 0.3 | 0.4 | m |
| SIE* | Arctic September sea-ice extent | 4 | 3 | 2 | 1 | $10^6$ km$^2$ |

| ΔAMOC | Change in mean annual strength of the AMOC relative to 1850–1900 | -20 | -25 | -30 | -40 | % |
|---|---|---|---|---|---|---|
| $A_{SO}$ | Mean annual area proportion of Southern Ocean surface waters (south of 50°S) with aragonite undersaturation ($\Omega_{arag}<1$) | 20 | 40 | 60 | 80 | % |
| $A_{Arctic}$ | Mean annual area proportion of Arctic Ocean surface waters (north of 70°N) with aragonite undersaturation ($\Omega_{arag}<1$) | 20 | 40 | 60 | 80 | % |
| $A_{\Omega<3}$ | Mean annual area proportion of global ocean surface waters with $\Omega_{arag}<3$ | 50 | 70 | 90 | 100 | % |
| Hypoxic $\Delta O_2$ | Change in mean annual volume of hypoxic waters (<63 umol L$^{-1}$) above 1000 m relative to 1850–1900 | 2 | 4 | 6 | 8 | % |
| Global $\Delta O_2$ | Change in mean annual global $O_2$ content relative to 1850–1900 | -1.8 | -2.4 | -2.6 | -3.5 | % |
| ΔNPP* | Change in mean annual depth-integrated net primary production relative to 1850–1900 | -2 | -3.5 | -4 | -8 | % |
| ΔBiomass* | Change in mean annual depth-integrated plankton biomass relative to 1850–1900 | -2 | -3.5 | -4 | -8 | % |
| ΔΦ | Change in mean annual upper-ocean (depth < 400 m) metabolic index relative to 1850–1900 | -5 | -10 | -15 | -20 | % |
| ΔPOM | Change in mean annual particulate organic matter flux at 100 m depth (30°N–20°S) relative to 1850–1900 | -4 | -6 | -8 | -10 | % |

## 2.2 CMIP6 Earth system model ensemble

Our CMIP6 model ensemble is composed of 9 ESMs (Table 2). This ensemble is based on the one used in Canadell et al. (2021), but excluding model family duplicates, and using the variant r1i1p1f1 (or equivalent). All ESMs use ocean components with a nominal horizontal resolution of about 1° with grid refinements of up to about 1/3° both poleward and at the equator. We use 3 scenarios from CMIP6 covering the period 2015–2100, which are initialized from the end of the historical simulation (1850–2014) that is based on estimates of historical forcings (O'Neill et al., 2016). These scenarios cover very different

possible futures: The low-emission, high-mitigation scenario SSP1-2.6 assumes that the world gradually shifts toward a more sustainable pathway, and that early and consistent climate mitigation limits the end-of-century radiative forcing to 2.6 W m$^{-2}$.

In contrast, the SSP5-8.5 scenario assumes resource-intensive, strong economic growth based on the exploitation of fossil fuel reserves and no climate mitigation. The very high carbon dioxide ($CO_2$) emissions in this scenario lead to a radiative forcing of 8.5 W m$^{-2}$ at the end of this century. The SSP5-3.4-OS scenario follows the SSP5-8.5 pathway up to year 2040. Then, strong climate mitigation policies are implemented, including carbon dioxide removal from the atmosphere, leading to a peak and decline in surface temperature and a final radiative forcing level of 3.4 W m$^{-2}$ in 2100. To use the same model ensemble for

all scenarios, we excluded models that do not provide SSP5-3.4-OS.

**Table 2. CMIP6 ensemble and variable availability per model.**

| Model | Reference | Variable availability |
|---|---|---|
| ACCESS-ESM1-5 | Ziehn et al. (2020) | SAT, SIE, $O_2$, $\Phi$, MHW, AMOC, $\Omega_{arag}$, SSL, POM, NPP |
| CanESM5 | Swart et al. (2019) | SAT, SIE, $O_2$, $\Phi$, MHW, Biomass, AMOC, $\Omega_{arag}$, SSL, POM |
| CESM2-WACCM | Danabasoglu et al. (2020) | SAT, SIE, Biomass, AMOC, $\Omega_{arag}$, POM, NPP |
| CMCC-ESM2 | Cherchi et al. (2019) | SAT, $O_2$, $\Phi$, MHW, Biomass, $\Omega_{arag}$, SSL, POM, NPP |
| CNRM-ESM2-1 | Séférian et al. (2019) | SAT, SIE, $O_2$, $\Phi$, MHW, AMOC, $\Omega_{arag}$, SSL, POM, NPP |
| IPSL-CM6A-LR | Boucher et al. (2020) | SAT, SIE, $O_2$, $\Phi$, Biomass, AMOC, $\Omega_{arag}$, SSL, POM, NPP |
| MIROC-ES2L | Hajima et al. (2020) | SAT, SIE, $O_2$, $\Phi$, Biomass, AMOC, $\Omega_{arag}$, POM, NPP |
| NorESM2-LM | Seland et al. (2020) | SAT, SIE, $O_2$, $\Phi$, MHW, Biomass, AMOC, $\Omega_{arag}$, SSL, POM, NPP |
| UKESM1-0-LL | Sellar et al. (2019) | SAT, SIE, $O_2$, $\Phi$, MHW, Biomass, $\Omega_{arag}$, SSL, POM, NPP |

All ESM model outputs used in this study have been regridded to a 1°-resolution regular grid (360x180 grid cells) before

analysis. Since most of the impact metrics are expressed as a change relative to the period 1850–1900, model biases would only be an issue for the analysis presented here if the response to forcing would significantly depend on the baseline state. However, we removed potential model drifts from two sensitive metrics ($\Delta$SSL and Global $\Delta O_2$) by calculating the difference between the projected signal and its equivalent from the corresponding preindustrial experiment (piControl). To account for carbonate chemistry biases in the present-day mean state simulated by the CMIP6 ESMs, we follow the methodology of

Terhaar et al. (2020). Changes in aragonite saturation state ($\Omega_{arag}$) have been computed offline using mocsy 2.0 (Orr and Epitalon, 2015) from regridded annual CMIP6 model outputs of dissolved inorganic carbon (DIC), alkalinity, sea water temperature (T), and sea water salinity (S). These modelled changes have then been added to the contemporary saturation state that we derived from the observation-based GLODAPv2 data product for DIC and alkalinity (Lauvset et al., 2016), and the World Ocean Atlas 2013 for T and S (Locarnini et al., 2013; Zweng et al., 2013).


For all impact metrics, time series have been smoothed using a 20-year running mean before identifying the years and global warming levels when a certain limit is exceeded. The exceedance is identified by the time and global warming level at which a given limit is exceeded for the first time. This definition does not account for "overshooting" limits of an impact metric. Cases where a limit is first exceeded, but the system returns to a state below the limit later in time, is counted as an exceedance.
Nevertheless, we provide analysis that allows for identifying cases where such overshooting of limit happens (Figs. 3 and 4).

Some limits might be exceeded by only a part of all available models or ensemble members, while other limits might be exceeded by all models or ensemble members within the time horizon of the scenario simulations (until 2100). If a model does not exceed a given limit, we interpret this as an exceedance in the last year of the simulation (year 2100). This approach is
conservative in the sense that it assigns the earliest possible exceedance year (the unknown true exceedance year of this model is later, or the model might not exceed the limit at all), and it ensures that all information provided by the model ensemble is used. A similar approach is applied along the global warming dimension, by attributing the highest warming level reached by the model under the given scenario to models not exceeding a given limit. Consequently, our exceedance estimates are characterised in terms of uncertainty and probability. We define exceedance uncertainty as the interquartile range of an
exceedance estimate in a given model ensemble for a given experiment, impact metric, and limit. We define exceedance probability as the proportion of models exceeding a limit across all models that provide data for a given impact metric and limit (i.e., here we exclude models that did not exceed the limit within the simulation period). We assign high probability to an estimate of exceedance time (or warming level) if a limit is exceeded by at least 80 % of our CMIP6 ensemble (i.e., at least 8 out of 9 ESMs) or of the ensemble members of an individual EMIC, medium probability if 50–79 % of the models or
ensemble members exceed the limit, and low probability if less than 50 % of models or ensemble members exceed the limit. This approach avoids the use of ensemble means, and uncertainty intervals such as ensemble standard deviation to define exceedances, from which a lot of information on the distribution of exceedances within the ensemble is lost leading to an underestimation of the resulting exceedance uncertainty. We will further discuss this issue below with some examples. We note that there are small differences as to which CMIP6 ESMs can be used for which metric, since not all models provide all
data necessary for all impact metrics (Table 2).

**2.3 Large perturbed-parameter ensembles from two Earth System Models of Intermediate Complexity**

In addition to output from CMIP6 ESMs, we also analyse scenario simulations of two Earth system models of intermediate complexity (EMICs), the Bern3D-LPX model and the University of Victoria Earth System Climate Model (UVic).
Both EMICs have simulated large perturbed-parameter ensembles (PPE) to estimate the range of parametric (model)
uncertainty. The EMIC PPEs were run over the historical period as well as for the SSP1-2.6, SSP5-3.4-OS, and SSP5-8.5 scenarios. Ensemble generation, sampled parameters, and calculation of ensemble member skill scores based on observations differ between the two models and are briefly outlined below.
**Bern3D-LPX model**

The model setup, ensemble generation, and evaluation as well as the experimental protocol of the Bern3D-LPX ensemble are the same as detailed in Jeltsch-Thömmes et al. (2024). The model features a three-dimensional dynamic ocean (Edwards et al., 1998; Müller et al., 2006) including sea-ice, a single-layer energy and moisture balance model of the atmosphere (Ritz et al., 2011), and a comprehensive terrestrial biosphere component (LPX-Bern v1.5) with dynamic vegetation, fire, nitrogen, nitrous oxide, methane, permafrost, peatland, and land-use modules (Lienert and Joos, 2018).

The sampling approach for the PPE builds upon work by Steinacher et al. (2013) and has been used thereafter in several follow-up studies (e.g., Steinacher and Joos, 2016; Battaglia et al., 2016; Battaglia and Joos, 2018; Lienert and Joos, 2018). A 1000-member PPE is generated from the prior distributions of 27 key model parameters using Latin hypercube sampling (Mckay et al., 2000; Steinacher et al., 2013).

To reduce uncertainties, we exploit a broad set of observation-based data (Fig. A1) to constrain the model ensemble to realisations that are compatible with observations, thereby probing both the mean state and the transient response in space and time of the ensemble members. Further details on the methods used to constrain the Bern3D-LPX model ensemble with observations are provided in Jeltsch-Thömmes et al. (2024).

**UVic ESCM v2.10**

The UVic ESCM v2.10 (Mengis et al., 2020; Weaver et al., 2001) has a three dimensional ocean with a horizontal resolution of 3.6° longitude, 1.8° latitude, and 19 vertical levels. The atmosphere is represented by a two-dimensional energy-moisture balance model with the same horizontal resolution (Fanning and Weaver, 1996). The oceanic physics follows the Modular Ocean Model version 2 (MOM2) (Pacanowski, 1996) and the ocean biogeochemistry model is outlined by (Keller et al., 2012). A thermodynamic-dynamic sea ice model (Bitz et al., 2001) employing elastic visco-plastic rheology (Hunke and Dukowicz, 1997) is coupled to the ocean. The terrestrial component accounts for vegetation dynamics and incorporates five different plant functional types (Meissner et al., 2003). Additionally, the model includes a representation of permafrost carbon (MacDougall et al., 2017) using a diffusion-based scheme, which approximates the process of cryoturbation.

We adopt a similar approach as described in Jeltsch-Thömmes et al. (2024) for the Bern3D-LPX model to generate the PPE and compute PPE member's scores based on observations. An emulated ensemble of 1978 members was weighted using this score and used for all statistical computations in this work (see appendix A for details).

## 3 Results and Discussion

### 3.1 CMIP6 ESM

The uncertainty, and probability related to the time and global warming levels at which limits are exceeded are highly variable across impact metrics, limits, and scenarios (Figs. 1 and 2). The probability in exceedance estimates generally decreases with higher limits, since generally less models exceed the higher limits. Note that the interquartile ranges in Fig. 1 are extended toward or up to 2100 in case that only some of the ESMs exceed a given limit, because of our choice to assign the year 2100 as time of exceedance for those models. All the reported median exceedances related to the most ambitious limit (limit 1) are

projected to occur in the short and mid term (before 2060) for all scenarios. If limit 1 is exceeded with high probability (marked by full size black dots in Figs. 1 and 2) in all scenarios, the median time of exceedance is generally very similar across scenarios ($\Delta$SAT, $\Delta$SSL, SIE, Arctic Ocean $\Omega_a<1$, $\Delta$Biomass ), because during earlier times the three scenarios share the same historical forcing or have only slightly diverged. Also, before 2040, the two scenarios of the SSP5 family are identical by construction, such that if all models exceed a limit before 2040, the median exceedance time is identical for SSP5-3.4-OS and SSP5-8.5. Metrics related to surface ocean aragonite saturation state (Arctic Ocean $\Omega_a<1$, Southern Ocean $\Omega_a<1$ and Global Ocean $\Omega_a<3$) show particularly narrow uncertainties over the time dimension, but not over the global warming levels. This is consistent with the findings of Terhaar et al. (2023) showing that projections with prescribed atmospheric $CO_2$ yield an unrealistic small uncertainty for surface ocean acidification, because the forcing agent (atmospheric $CO_2$) is the same across all ensemble members. If projections target a certain temperature level, a more relevant estimate of uncertainty can be obtained because variations of climate sensitivity across ensemble members translate into different levels of atmospheric $CO_2$ at the targeted temperature, and hence into different levels of surface ocean acidification. Lower exceedance probability and larger difference in the timing of exceedance are found for $\Delta$POM, $\Delta$NPP, global $\Delta$O2, and hypoxic $\Delta$O2. The lower probability in exceedance of $\Delta$NPP limits across all scenarios is consistent with the high uncertainty in $\Delta$NPP projections found by Kwiatkowski et al. (2020). In contrast, the higher exceedance probability linked to the metric $\Delta$Biomass is consistent with the findings of Tittensor et al. (2021), emphasising that Biomass is a more robust metric for assessing impacts of climate change on marine ecosystems than NPP. Substantial uncertainties in global $\Delta$O2 were found in earlier multi-model studies (Cocco et al., 2013; Hameau et al., 2020) , which are reflected by the lower probability of exceedance estimates. These uncertainties can be explained by the uncertain balance between $O_2$ supply from physical mixing and advection, and $O_2$ consumption from remineralization of organic matter. Uncertainties remain also on how these processes would respond to rising $CO_2$. Regarding subsurface $O_2$ projections, Frölicher et al. (2016) identified model structure and parametrization as the second source of projection uncertainty, consistent with the results presented in Figs 1 and 2.

The 20-year moving averaging applied to the time series mostly removes interannual to decadal variability and emphasises the signal induced by the different radiative forcing. However, metrics with large internal variability, like the AMOC strength, can still show exceedance distribution not entirely consistent with the different radiative forcing levels applied in the emissions scenarios (Figure 1, limit 2). Similarly, composite metrics such as $\Delta$Biomass (i.e. sum of phytoplankton and zooplankton biomass) include interactive dynamic responses that lead to inconsistent responses with respect to the different radiative forcing levels for some models with exceedances only 2 to 3 years earlier in SSP1-2.6 compared to SSP5-8.5. Hence, differences between scenarios in exceedance timing of around 10 years should not be considered significantly different (i.e., they can be considered as occurring simultaneously).

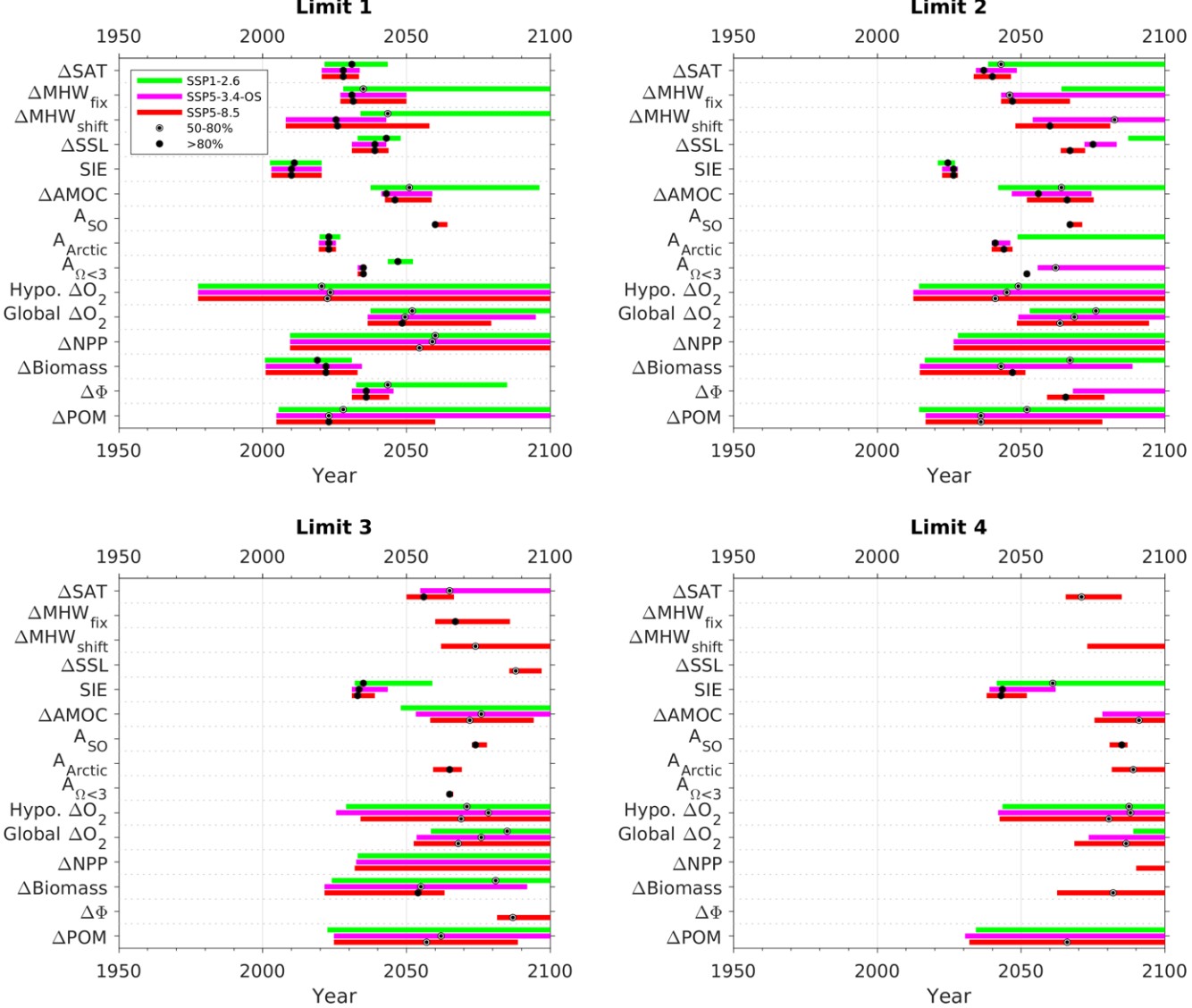

**Figure 1: Box plots showing the distribution of exceedance years for each limits of the impact metrics (abbreviations follow Table 1) for CMIP6 models. Green, purple and red colors depict the three scenarios, i.e., historical-SSP1-2.6, historical–SSP5-3.4-OS and historical–SSP5-8.5. Boxes' lengths and circled black dots depict the interquartile range and the ensemble median, respectively. The size of the black dots indicates the exceedance probability defined as the proportion of models exceeding a limit across the models providing data for a given impact metric (Full size: ≥ 80 %, half size: 50–79 %). The median of low probability exceedances (<50 % of models) are not shown.**

350

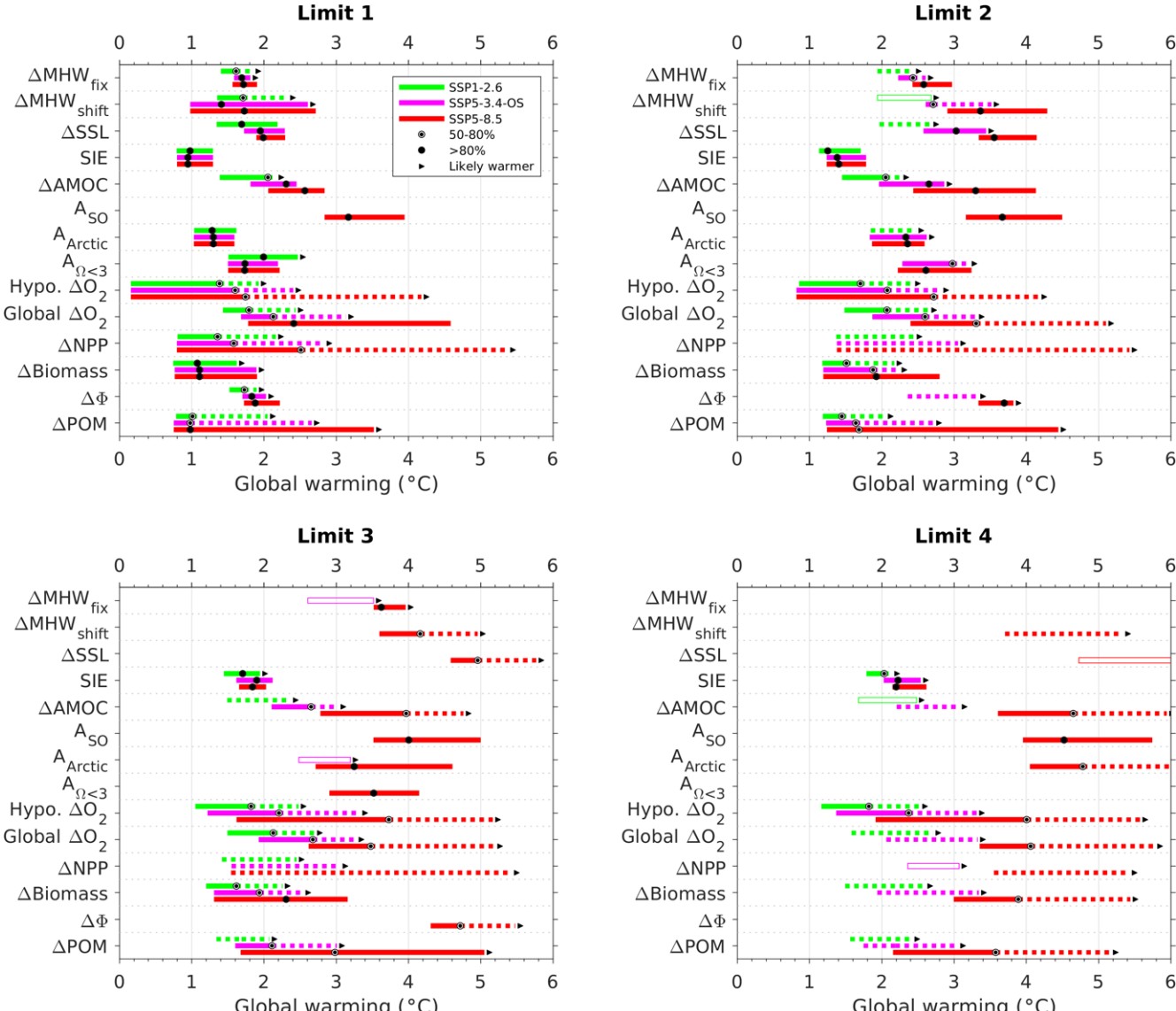

**Figure 2: Same as Figure 1, but relative to global warming in degree Celsius during this century instead of time. Cases with very low probability (<25 %) are displayed as empty rectangles, using the interquartile range as in Figure 1. Median symbols are not shown for these cases. Low probability cases in the 25–50 % probability range are represented by dashed bars covering the interquartile range, without median symbols either. Black arrows pointing toward warmer levels emphasize that the limit would likely be exceeded only at warmer global warming levels. Such arrows are added to all cases in the 50–100 % probability range (100 % excluded). Medium probability cases with 50–79 % have interquartile ranges shown as plain bars with median symbols, and dashed bars over the 50th– 75th percentile range to highlight the uncertainty of the 50th–75th percentile range. Finally, cases with probability ≥ 80 % follow Figure 1's legend (plain bars over interquartile range with median symbol).**

For the less ambitious limits, exceedance time estimates generally move towards later times and higher warming levels, and the exceedance probability decreases (less models exceed the less ambitious limits). In the low emission scenario SSP1-2.6, limit 4 is exceeded by less than 20 % of the CMIP6 ESMs for any of the impact metrics due to the stringent climate mitigation

implemented in this scenario. Exceedances occur generally at lower global warming level under scenarios with lower radiative forcing (Fig. 2). For high probability cases, this is due to the inherent inertia from some metrics (e.g., Fig. 2, limit 1, $\Delta$SSL) that eventually leads to an exceedance even under the global warming stabilization induced by SSP1-2.6 scenario. For low

370 probability cases, this observation is only the result of the default attribution of the highest warming level reached to models not exceeding a given limit (Fig. 2, limit 4, $\Delta$Biomass).

If we focus on exceedance estimates that have a high probability (i.e., where at least 80 % of models exceed the limit), we can provide an estimate of the time when limits are likely to be exceeded (Figs. 3 to 6). The first limit of Arctic Ocean $\Omega_a$<1 (20 % of undersatured surface waters) is likely already passed in the CMIP6 model ensemble in all scenarios, consistent with the

375 findings of Terhaar et al. (2021). The first limit of SIE (4 $10^6$ km$^2$) is expected to be passed during 2013–2034 (median year 2023). So far, according to satellite-based estimates, Arctic summer sea-ice extent fell below 4 $10^6$ km$^2$ in 2012 and 2020 (https://nsidc.org/, visited on May 21[st], 2025). The fourth level of limits is exceeded with high probability only for SIE in SSP5-3.4-OS and SSP5-8.5, as well as for Southern Ocean $\Omega_a$<1 in SSP5-8.5.

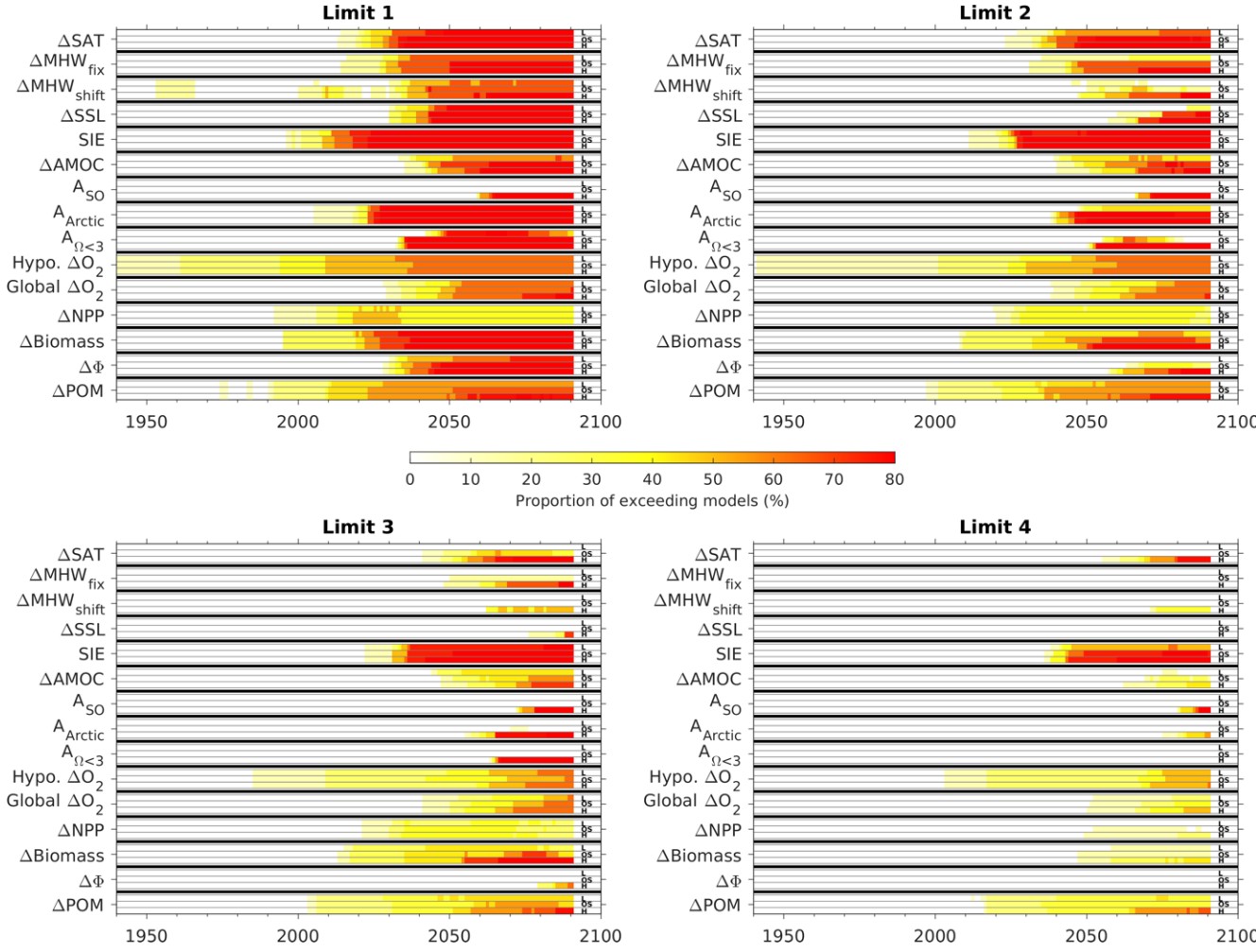

**Figure 3: Proportion of models exceeding a given limit of the impact metrics compared to the available models for each metric. Abbreviations are included for SSP1-2.6 (L), SSP5-3.4-OS (OS) and SSP5-8.5 (H).**

Avoiding emissions as high as in the SSP5-8.5 scenario and following an emission pathway similar to SSP5-3.4-OS will likely avoid an exceedance of any of the limits for $\Delta MHW_{shift}$, Southern Ocean $\Omega_a < 1$, Global $\Delta O_2$, and $\Delta POM$ during this century.
In addition, avoiding the SSP5-3.4-OS scenario by early mitigation (as in SSP1-2.6) will likely avoid exceedances of any limit related to $\Delta MHW_{fix}$, $\Delta AMOC$, $\Delta Biomass$, and $\Delta \Phi$ until year 2100. The limits of $\Delta SAT$, $\Delta SSL$, SIE, Arctic Ocean $\Omega_a < 1$, $A_{\Omega < 3}$, and $\Delta Biomass$ are likely to be exceeded across all scenarios. The effect of ambitious mitigation in SSP5-3.4-OS after 2040 can be seen for $\Delta SAT$, $\Delta SSL$, SIE, and $\Delta AMOC$. For these metrics, some of the limits are first exceeded around mid-century, but later this exceedance is reversed. The uncertainty of hypoxic $\Delta O_2$, global $\Delta O_2$, $\Delta NPP$, and $\Delta POM$ projections does not allow us to conclude with confidence that none of the corresponding limits will be exceeded due to their attributed low probability (Fig. 4).

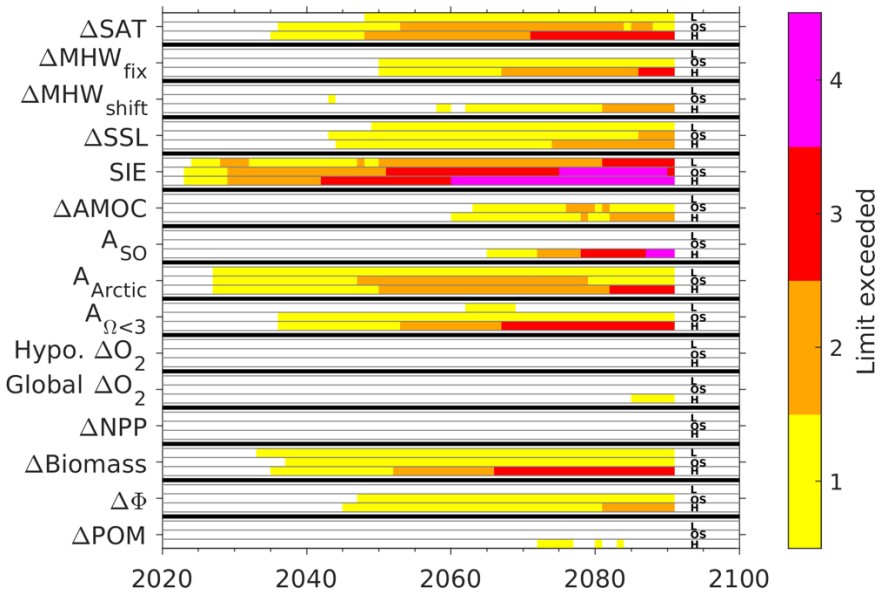

**Figure 4: Time periods where limits are exceeded with high probability according to the CMIP6 model ensemble (≥ 80 % of models) for each impact metrics and for the scenarios SSP1-2.6 (L), SSP5-3.4-OS (OS) and SSP5-8.5 (H).**

In Figs. 5 and 6, the summary of high probability exceedance estimates for all impact metrics is quite conservative, since (1) high probability is defined as at least 80 % of models exceeding a limit (i.e., at least 8 out 9 models, which is practically 88 %) and (2) medium probability exceedances (where up to 79 % of the CMIP6 models would show an exceedance of a given limit) are not included. For the low-emission scenario, already the most ambitious level of limits of 5 impact metrics ($\Delta$SAT, $\Delta$SSL, SIE, Arctic Ocean $\Omega_a$<1, and $\Delta$Biomass) is exceeded with high probability, two of them as early as in the near-term period (2021-2040). In contrast, even the third and fourth set of limits are exceeded with high probability in the high-emission SSP5-8.5 scenario, particularly toward 2100. The absence of high probability in the exceedance for hypoxic $\Delta O_2$, global $\Delta O_2$, $\Delta$NPP, and $\Delta$POM is explained partly by a model disagreement within our CMIP6 ensemble. Another reason for low-to-medium probability in the exceedance for global and hypoxic $\Delta O_2$ and $\Delta$SSL before year 2100 is that changes in subsurface and whole ocean parameters have been shown to accrue beyond year 2100 and aggravate over many centuries due to the long overturning time scales of the ocean (Battaglia and Joos, 2018). There is a very clear effect of the ambitious mitigation assumed in SSP5-3.4-OS after 2040 in all timeseries of impact metrics, such that the significantly lower exceedance rate of limits, particularly by the end of the century, clearly illustrated the benefits of stringent and ambitious mitigation. Some metrics show strong hysteresis in response to cumulative carbon emissions (Boucher et al., 2012; Jeltsch-Thömmes et al., 2020; Samanta et al., 2010; Santana-Falcón et al., 2023). Due to that hysteresis behavior, sustained negative emissions are required to return to and stay under a specific limit, particularly for high climate sensitivities and peak-and-decline scenarios with carbon dioxide removal (Jeltsch-Thömmes et al., 2020). This aspect needs to be emphasized in the case of our study due to the use of

simulations ending in 2100, implying that some metrics could return below more ambitious limits beyond 2100 under strong
415   mitigation scenarios.

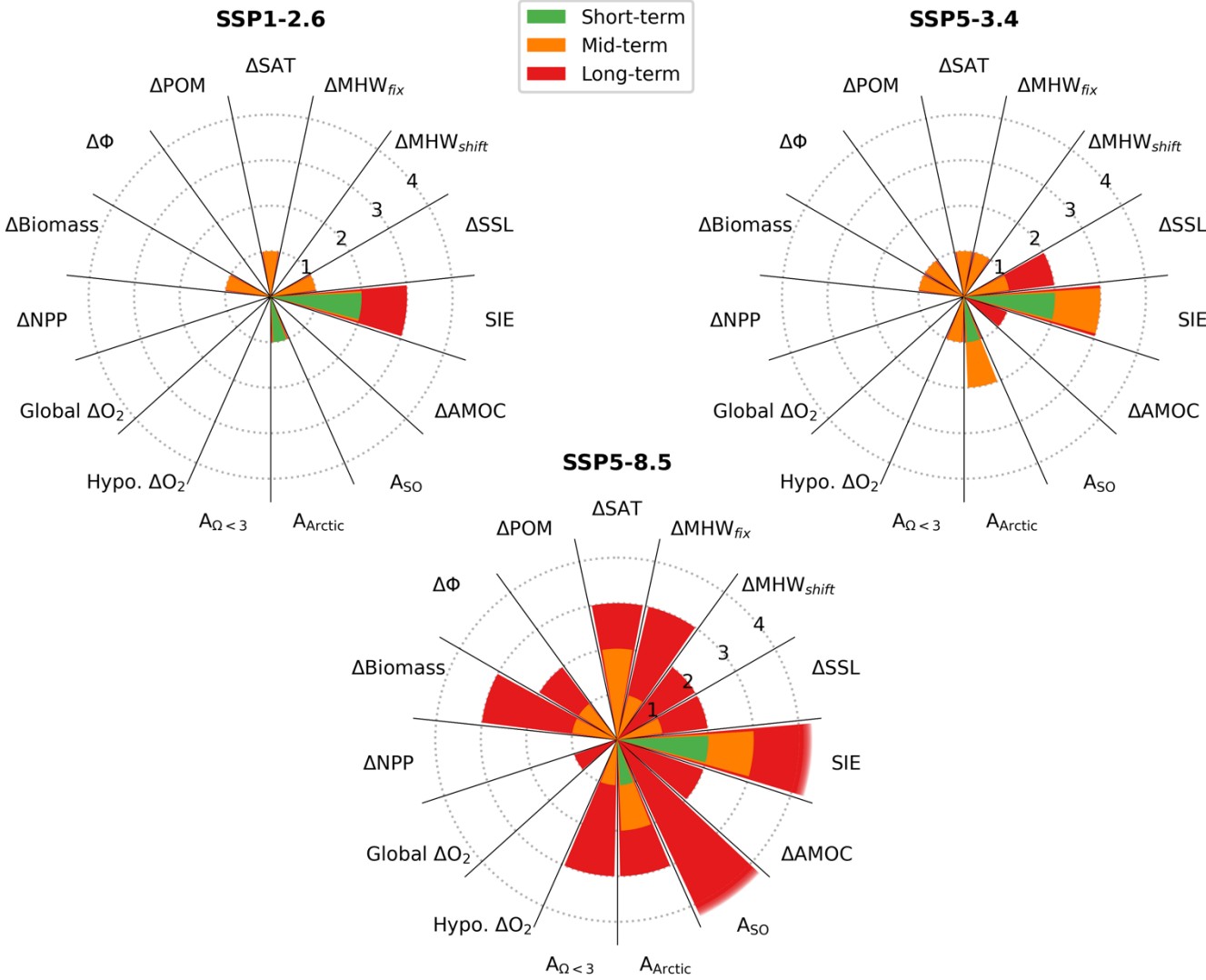

**Figure 5: Exceedance of limits with high probability (≥ 80 % of the CMIP6 models) in the near-term (2021–2040), mid-term (2041–2060) and long-term (2081–2100) periods under (left) SSP1-2.6, (right) SSP5-3.4-OS and (middle) SSP5-8.5 scenarios.**

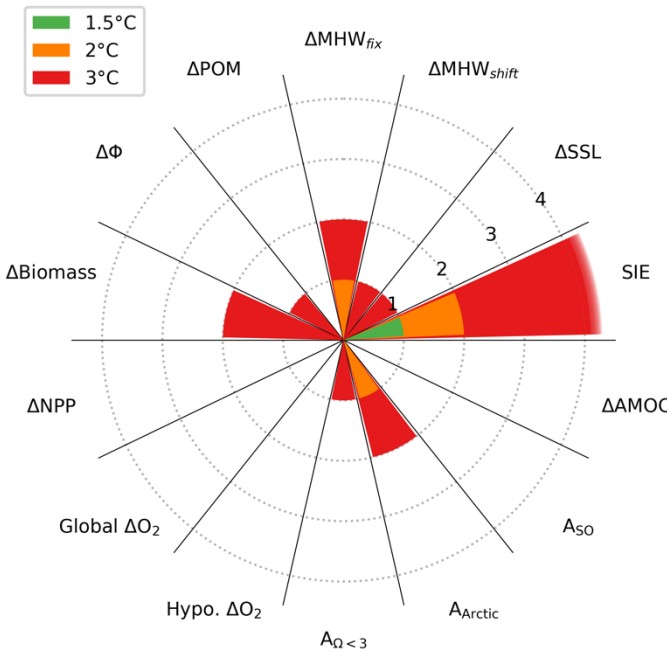

**Figure 6: Same as Figure 5, but according to global warming levels (1.5°C, 2°C and 3°C) under the SSP5-8.5 scenario. Therefore, ΔSAT is not shown.**

### 3.2 EMIC & ESM comparison

There is broad agreement for a range of variables on the exceedances of limits between the CMIP6 and the two skill-weighted EMIC ensembles under SSP5-8.5, but also major disagreements are identified (Figs. 7 and 8). Note that the median and interquartile ranges are only shown for ensembles that pass the limits with a probability of 50 % or more. For a good agreement between ensembles both probability and median needs to match. For the limit 2, the median value of exceedance agrees within 25 years and 1°C for ΔSAT, ΔAMOC, Southern Ocean $\Omega_a<1$, Arctic Ocean $\Omega_a<1$, Global Ocean $\Omega_a>3$, and ΔPOM between the CMIP6 and the two EMIC ensembles. However, probability is variable among ensembles. The two EMIC ensembles generally show systematically high probability in limit exceedances in the first two limit sets (except limits 2 of UVic's ΔΦ and Bern3D Hypoxic $\Delta O_2$) while the CMIP6 ensemble shows only 4 exceedances with medium probability over the same limits. The CMIP6 ensemble shows somewhat larger warming than the EMIC ensemble with earlier exceedance. This is consistent with the fact that the CMIP6 ensemble includes models with climate sensitivity larger than observation-constrained estimates (Nijsse et al., 2020; Tokarska et al., 2020). For the third and fourth limit sets, the CMIP6 models show medium

probability in the exceedance of the hypoxic $\Delta O_2$, whereas the EMIC ensembles show no exceedance or with little probability. The finer spatial resolution used in CMIP6 models compared to the EMIC ensemble could explain this difference.

Regarding the uncertainties related to the exceedance of limits, the three ensemble agrees generally well over many metrics (Fig. 7). Regarding $\Delta$POM and hypoxic $\Delta O_2$, the uncertainty range from the CMIP6 ensemble is larger than the EMIC ensembles. We hypothesize that the parametric uncertainty sampled in the perturbed-parameter EMIC ensembles is a significant underestimation of the full uncertainty signal of some metrics as it lacks the structural model uncertainty inherent in the CMIP6 ensemble. The ESMs' uncertainties of exceedances related to Southern Ocean $\Omega_a<1$, Arctic Ocean $\Omega_a<1$, Global Ocean $\Omega_a>3$ are narrower than the one from EMICs. We interpret this difference by an efficient bias correction of the present-day mean state applied to these metrics for CMIP6 ESMs. Another explanation could be how EMIC ensembles are constrained. For UVic, global mean profiles of ocean tracers have been used as observational constraints. The latter averages large regional variations that compensate each other resulting in similar global mean. Such globally averaged constraints might inefficiently reduce uncertainty for ice-dominated polar regions, especially since we did not constrain sea-ice area. Large variations in sea-ice cover could influence air-sea gas exchange and, as a result, Arctic Ocean $\Omega_a<1$.

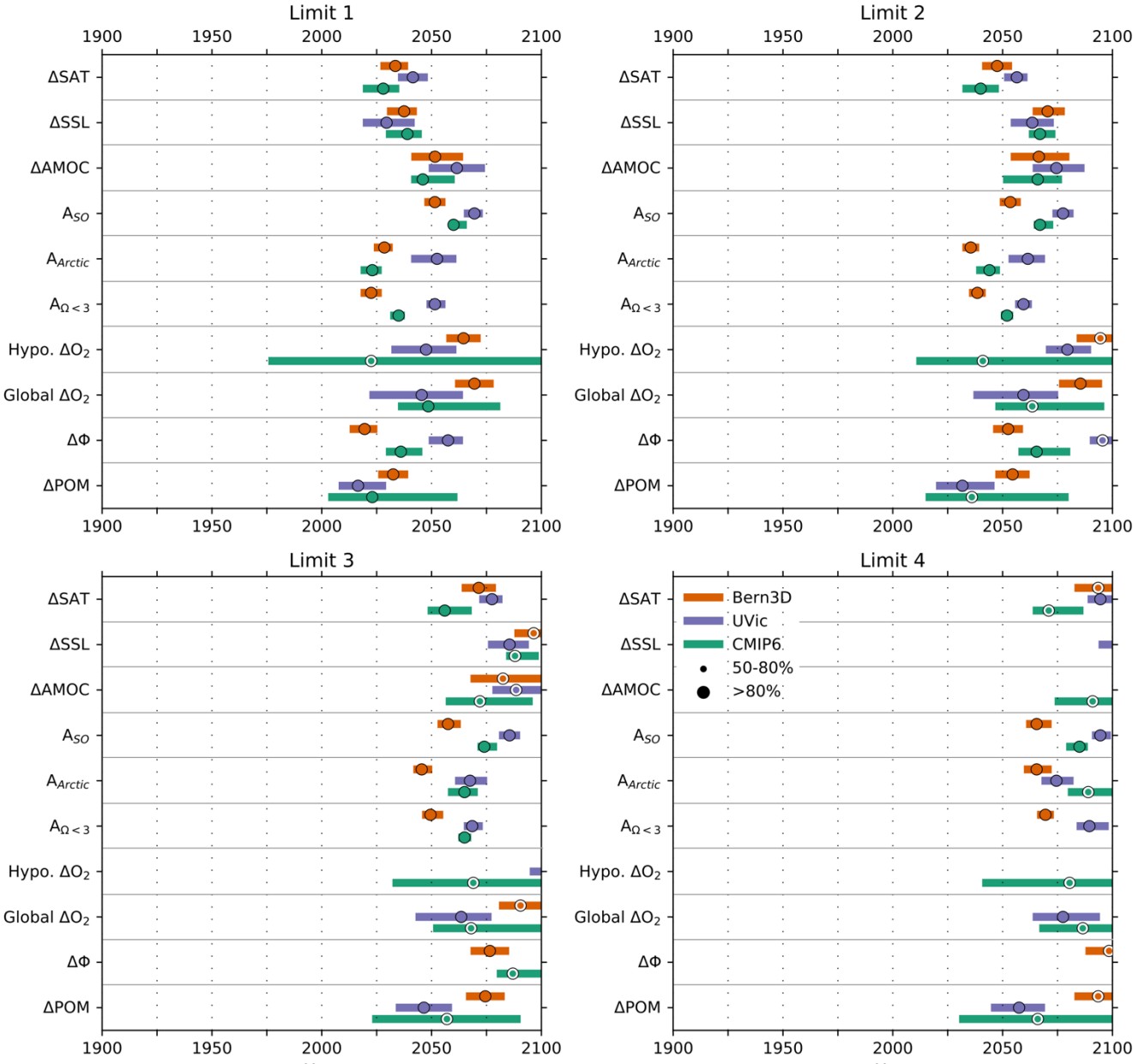

**Figure 7: Box plots showing the interquartile range distribution of exceedance years for each limit of the impact metrics (abbreviations follow Table 1) for SSP5-8.5. Orange, purple, and green show data from the Bern3D-LPX, UVic, and CMIP6 ensemble, respectively. Dots indicate the median and the size of the dots indicates the percentage of ensemble members that have crossed the respective limit. The median and interquartile ranges are only shown for ensembles that pass the limits with a probability of 50 % or more. MHW, SIE, NPP and ΔBiomass are not shown because EMIC ensembles were not able to provide data for these metrics.**

Despite such differences, robust conclusions emerge. First, both the CMIP6 and EMIC ensembles show medians passing most of the stringent limits of set 1 and set 2 with high probability within this century for global warming of 2.5°C and 3.5°C,

respectively (Fig. 8). Exceptions are ΔΦ, ΔSSL (known to lag surface warming and continues to increase over centuries), and ΔO₂ metrics, for which the CMIP6 and EMIC ensembles disagree. Second, both the CMIP6 and EMIC ensembles demonstrate that many less stringent limits of set 3 and set 4 are not passed with high probability within this century for global warming of 1.5°C and 2°C, respectively (Fig. 8). Taken together, the results of the model ensembles collectively show that limiting global warming below 2°C avoids passing the considered Earth system limits during this century with potentially dangerous impacts on eco- and socio-economic systems.

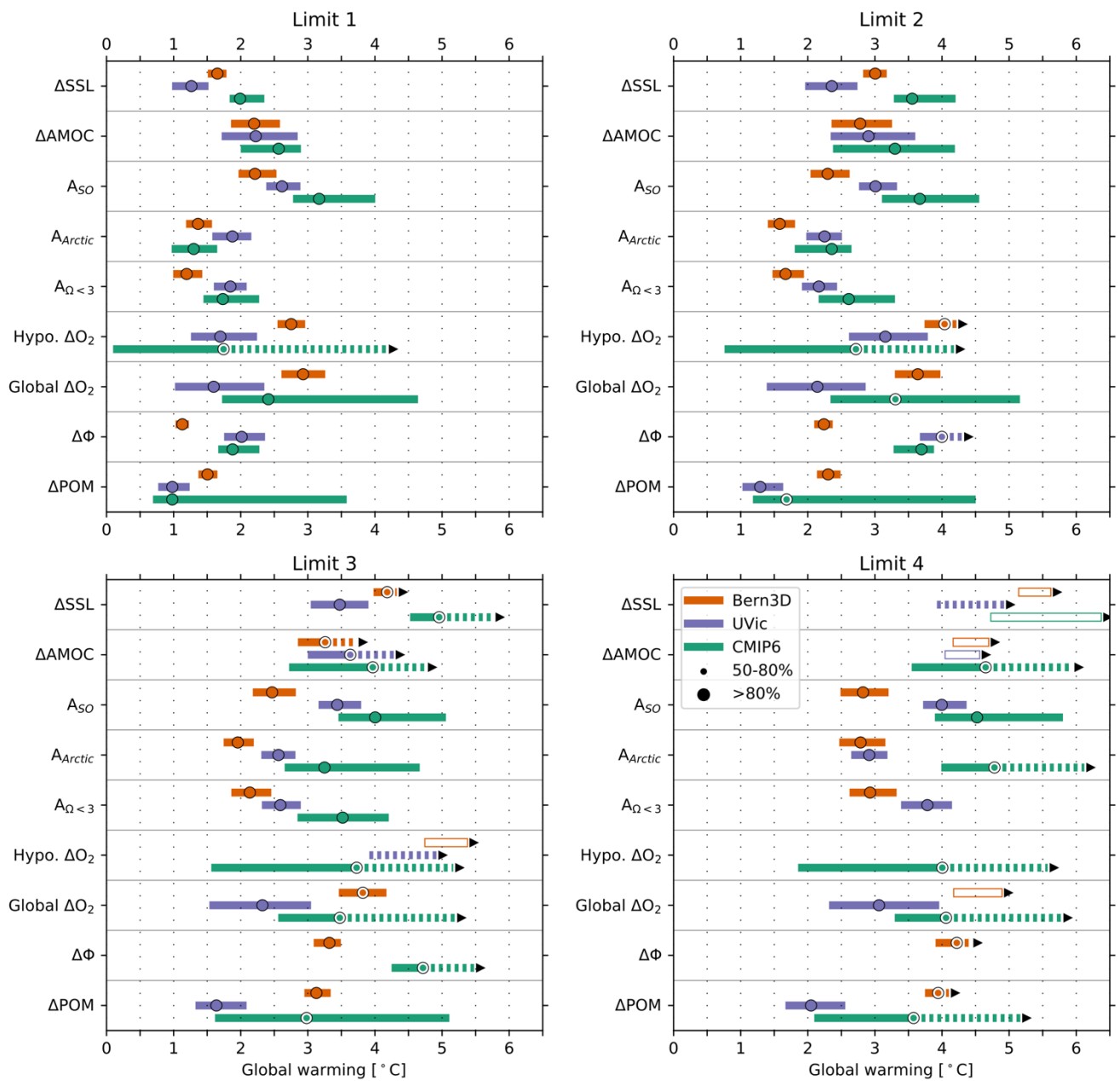


## 4 Implications and Conclusion

This study assesses different types of IPCC emission pathways (low, high, overshoot) with respect to multi-dimensional safe operating spaces quantified by a wide range of physical, chemical, and biological ocean impact metrics and corresponding limits based on the literature. It contributes to identifying viable mitigation pathways for the 21st century projected by state-of-

the-art Earth System Models, complemented by two observation-constrained ensembles from Earth System Models of Intermediate Complexity.

### 4.1 Limitations

Our study assesses a wide range of mitigation metrics over three very distinct scenario pathways and uses a large set of model results. Nevertheless, the global viewpoint that our study takes imposes certain limitations. Most important, the definitions of

our metrics use averages over the global ocean or large oceanic regions neglecting spatial heterogeneity of changes in the climate system. For instance, while our metric Arctic Ocean $\Omega_a<1$ averages across the whole ocean north of 70°N, there is significant regional variability in the extent and rate of aragonite undersaturation across the Arctic Ocean. Areas with high freshwater input from sea ice melt, such as the Canadian Arctic Archipelago, experience more rapid acidification compared to regions influenced by Atlantic inflow, which have greater vertical mixing and primary production (Popova et al., 2014).

Secondly, due to data limitations, we use only one ensemble member per ESM and scenario (many ESMs provide only one ensemble member). This is a limitation for metrics where changes in internal variability are important, as the marine heatwave metrics. For these metrics, the relative contributions of long-term warming trends versus anthropogenic changes in internal variability vary geographically, with the warming trend often accounting for more than 90 % of the total changes (Deser et al., 2024; Frölicher et al., 2018; Frölicher and Laufkötter, 2018). By using a single simulation per model, we can provide a general

assessment, but we fold together both model structural uncertainty and sampling uncertainty. To better assess MHW metrics under a moving-mean baseline, future studies should incorporate initial-condition large ensemble simulations (Burger et al., 2022; Deser et al., 2024), which would allow for a more robust evaluation of evolving forced responses in individual models at specific locations and times. Finally, for the metabolic index, we limited ourselves to illustrating the scenario- and model dependent changes in viable habitat based on metabolic traits for one median ecophysiotype (Fröb et al., 2024), while including

a broader range of species' thermal and hypoxia sensitivities would allow for a broader multi-species assessment of habitability.

### 4.2 Physical changes

With these limitations in mind, we can draw several conclusions from our results, firstly for physical metrics. For SSL rise, models agree relatively well on the timing of when certain limits are crossed. Also, the value of stringent mitigation is clearly

visible in our results, since the less ambitious limits are only breached in the high emission scenario within this century. This also illustrates the value of late mitigation (compared to no mitigation) in the overshoot scenario, because also here the crossing of the two least ambitious limits is avoided. This is consistent with previous work showing that the rate of SSL rise is quite reversible in overshoot scenarios (Schwinger et al., 2022). It should be stressed, however, that sea level rise has a high inertia and will continue beyond the end-of-century time-horizon considered in this study, even in the strong mitigation scenarios.

Our results highlight committed severe risks for marine ecosystems related to summer Arctic sea ice retreat whatever the emissions scenario. Even the least ambitious limit (ice-free Artic during summer; September sea-ice extent $< 10^6\,\mathrm{km^2}$) is passed with medium probability (50–79 % of models) in the low emission scenario. The summer Arctic sea ice reduction affects marine biota, particularly species like the Arctic cod, which are crucial for the diet of upper trophic level predators such as the black guillemot. This leads to changes in diet composition and increased nestling starvation rates among seabirds (Divoky et

al., 2015). The decline in Arctic sea ice directly threatens the food security and cultural continuity of Indigenous Peoples. Many Arctic communities rely on sea ice for hunting and fishing, which are essential for their livelihoods. The sea ice loss disrupts these traditional practices, leading to food insecurity and cultural disruption (Huntington et al., 2022). Further, the decline in sea ice extent is opening new shipping routes, such as the Northern Sea Route, which connects North-East Asia with North-Western Europe. This route reduces shipping distances by approximately one-third compared to the Southern Sea Route.

The opening of the Northern Sea Route could lead to significant economic benefits for global trade but raises geopolitical concerns and environmental pressures related to Arctic shipping and global supply chain reorganization (Bekkers et al., 2018). AMOC decline in CMIP6 models has been shown to be relatively insensitive to the emission scenario, at least up to 2060 (Weijer et al. 2020). This is also reflected in our results, since the median exceedances of limits 1 (20 % decline) and 2 (25 % decline) are not very different for the strong mitigation scenario SSP1-2.6 (albeit with a somewhat lower probability) compared

to SSP5-8.5 and SSP5-3.4. Nevertheless, the inter-model spread of projected AMOC decline remains large.

### 4.3 Ocean acidification

Results for the metrics related to ocean acidification show a high probability of crossing ambitious limits for the Arctic and the global ocean even in the low emission scenario SSP1-2.6 while the limits specific to the Southern Ocean are only crossed for the high emission scenario. Seen from a perspective of scenarios with prescribed atmospheric $CO_2$ concentration, the

uncertainty in these results is small. However, consistent with the study of Terhaar et al. (2023), uncertainties increase considerably when relating the ocean acidification related metrics to a specific global warming level. Ocean acidification poses a severe threat to early life stages of calcifying organisms which can dominate surface water communities in polar regions, e.g. the polar pteropod (Limacina helicina antarctica), leading to shell dissolution and fragility, high mortality, and reduced recruitment (Bednaršek et al., 2012; Gardner et al., 2018). This is of major importance as the pteropods contribute significantly

to the pelagic food web and carbon export fluxes in this region (Hauri et al., 2016). Aragonite undersaturation in the Arctic region threatens calcifying organisms such as plankton and invertebrates, which depend on calcium carbonate for their structural integrity. This can lead to changes in the composition of the Arctic ecosystem, affecting both planktonic and benthic

communities (Bates et al., 2013; Yamamoto-Kawai et al., 2009). At $\Omega<3$, the global ocean experiences widespread negative biological and ecological effects, including reduced survival, growth, and calcification in many marine species, especially those that build shells or skeletons from calcium carbonate (e.g., corals, molluscs, some plankton species; Kroeker et al., 2010). Together with long-term warming these can lead to declines in primary production and carbon export (Moore et al., 2018), resulting in lower fishery yields and reduced ecosystem productivity.

## 4.4 Biogeochemical and biological changes

Maybe not surprisingly, we find the largest uncertainties for the biogeochemical metrics such as the one related to $\underline{O_2}$, NPP, plankton biomass and POM. In all the scenarios used in our study, models do not agree on the projected sign of change of NPP while models agree on the projected decline of plankton biomass, albeit with significant uncertainties on the amplitude of this decline. These results are consistent with previous literature (Kwiatkowski et al., 2020; Tittensor et al., 2021). Change in NPP and biomass are influenced by complex interactions among nutrients, temperature, and ecosystem dynamics which is often beyond model capacity (Tagliabue et al., 2021). Regarding the metabolic index, viable habitat for marine organisms is lost if species-specific thresholds of metabolic demand and oxygen availability are crossed. Future projections show a decline in ocean habitability due to the combined threat of ocean warming and deoxygenation, leading to high extinction risk for polar species and loss of biological richness in the tropics (e.g., Penn and Deutsch, 2022).

## 4.5 Concluding remarks

Assessing the exceedance of limits for multiple impact metrics requires large model ensembles to obtain high-probability signals and corresponding uncertainties for the exceedance estimates (in years and global warming levels) linked to the projection pathways. Our analysis clearly indicates the need for better constraining and/or weighting the CMIP6 ensemble to reduce the large uncertainties found for exceedance estimates of many of the impact metrics. Simulations beyond year 2100 are needed to assess the long-term impacts of anthropogenic emissions, especially for the volume of hypoxic waters, global oxygen inventory, AMOC response, and steric sea level rise.

Our results show that ambitious limits will be exceeded with high or medium probability even if a low-emission pathway is followed, but that exceeding less ambitious limits associated with a higher risk for severe impacts is unlikely in a low-emission scenario. In contrast, under the high-emission scenario, many of the less ambitious and more risk-prone limits are exceeded with high to medium probability. The benefit of strong mitigation efforts in the overshoot pathway is clearly measurable as a decrease in the exceedance probability of the least ambitious and most risk-prone limits. Nevertheless, our analysis clearly indicates a risk of more severe impacts in the overshoot scenario compared to the strong mitigation scenario, particularly in the mid-term, highlighting the benefit of early mitigation strategies to avoid an overshoot scenario.

**Appendix**

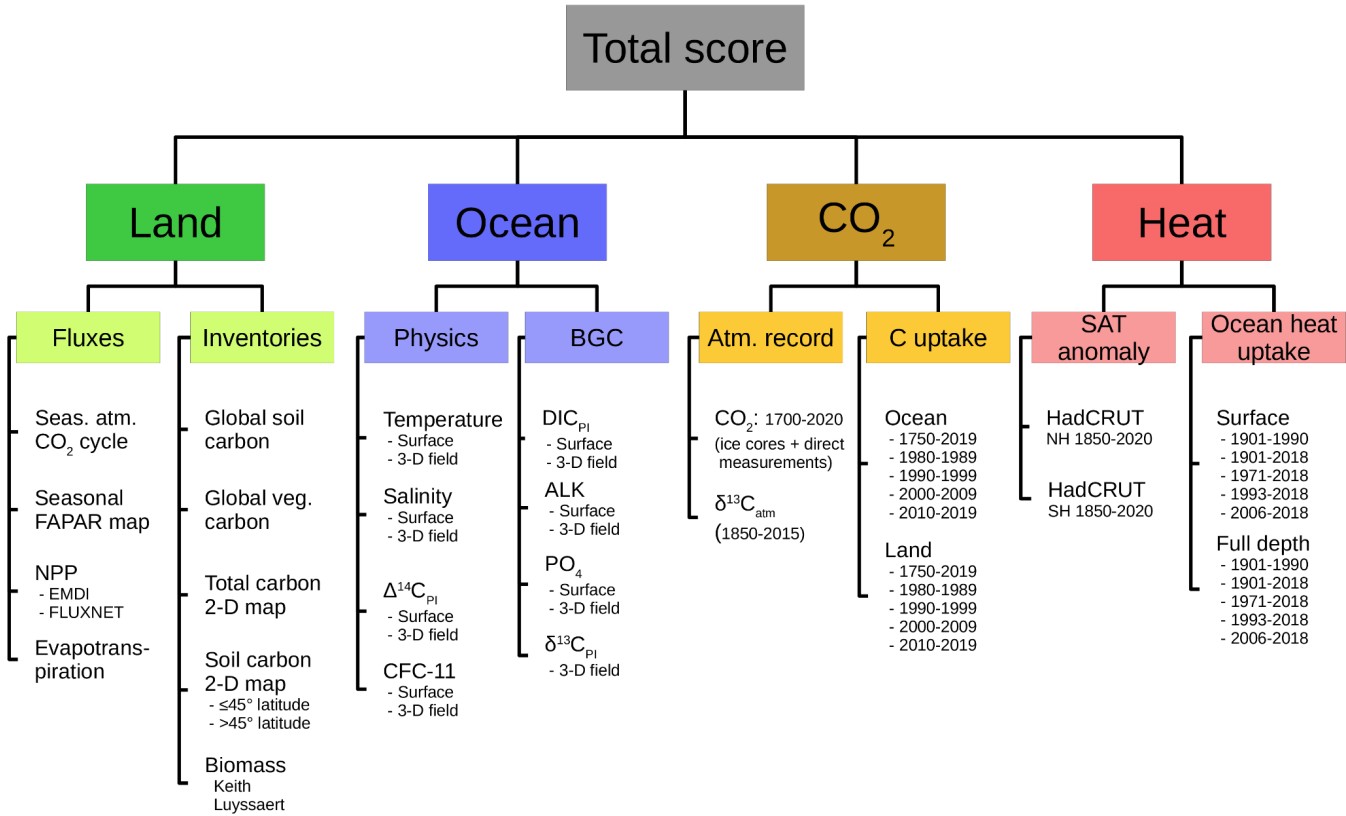

**Figure A1: Hierarchical weighting scheme used to calculate the skill scores of individual ensemble members of the Bern3D-LPX**
**model ensemble. Data sets at each level have equal weight. For example, the data-model mismatch in "Surface DIC" in the entry Ocean - BGC enters the total skill with a weight of 1/64 (½ (two data sets: Surface and 3-D field) x ¼ (4 groups: DIC, ALK, PO4, 13C) x ½ (two major subgroups: Physics, BGC) x ¼ (4 major groups: Land, Ocean, $CO_2$, Heat)).**

**UVic ESCM detailed methodology**

A 325-member PPE is generated using a multi-wave history matching approach (Andrianakis et al., 2015; Bower et al., 2010). History matching (HM), or iterative refocusing, is based on running an ensemble in a predefined parameter space, using it to train statistical emulators that predict key metrics from the model output, and then using the emulator to identify the set of inputs that would give an acceptable match between the model output and the observed data. In our case, we performed six waves of 80 simulations each and compared model outputs with observations after each wave. Gaussian Process (GP)

emulators (Kennedy and O'Hagan, 2001; Rasmussen and Williams, 2005; Sacks et al., 1989) are then constructed to predict these outputs as functions of the perturbed parameters to reject regions of the input space which are unlikely to produce results consistent with observations. For each quantity that we compare to observation, an implausibility measure (Andrianakis et al., 2015; Williamson et al., 2015) is computed following Eq. (1):

$$I_j(x) = \frac{|z_j - E^*[g_j(x)]|}{[V_o + V_c(x) + V_m]^{1/2}},$$ (1)

where $g_j(x)$ is the function describing the relationship between a vector of model inputs $x$ and a specific model output $j$. Since we employ GP emulation, we have the expectation provided by the emulators $E^*[g_j(x)]$. The corresponding observation is $z_j$. The term $V_o$, $V_c(x)$, and $V_m$ represent the variance associated with the observational uncertainty, the code uncertainty as given by the emulator, and the model discrepancy. The latter is simply defined as 10 % of the ensemble range due to the difficulty to estimate model discrepancy. The value of $I$ is large if it is unlikely for the model to produce an acceptable match with

observation when using the input combination $x$. We adopt a similar approach as described in Jeltsch-Thömmes et al. (2024) for the Bern3D-LPX model to compute a score $S$ based on our calculated implausibility measure $I_j(x)$. We generate a large Latin hypercube sampling plan and reject parameter combinations with emulated $I_j(x) > 3$. The emulated 1978-member ensemble was weighted using the score $S$ and used for all statistical computations in this work.

To assess the quality of our GP emulators, we employ leave-one-out cross-validation. This validation is conducted using the surface air temperature anomaly (ΔSAT) relative to the preindustrial period, spanning the years 2015 to 2100. For our ensemble of 325 simulations, we iteratively exclude one simulation and construct an emulator of ΔSAT using the remaining 324 simulations. The emulator is then used to predict the ΔSAT of the omitted simulation. This process is repeated for each simulation in the ensemble, ensuring that every simulation is excluded once. Additionally, validation is performed at every 5[th]

timestep in the time series.

Figure A2 presents an example of the emulated versus simulated ΔSAT values at every 5-time steps for a randomly selected ensemble member. The error bars represent the two standard deviations of the predicted mean estimated by the emulator. For a well-calibrated emulator, approximately 95 % of the true or simulated values should fall within 2 standard deviations of the emulated values.


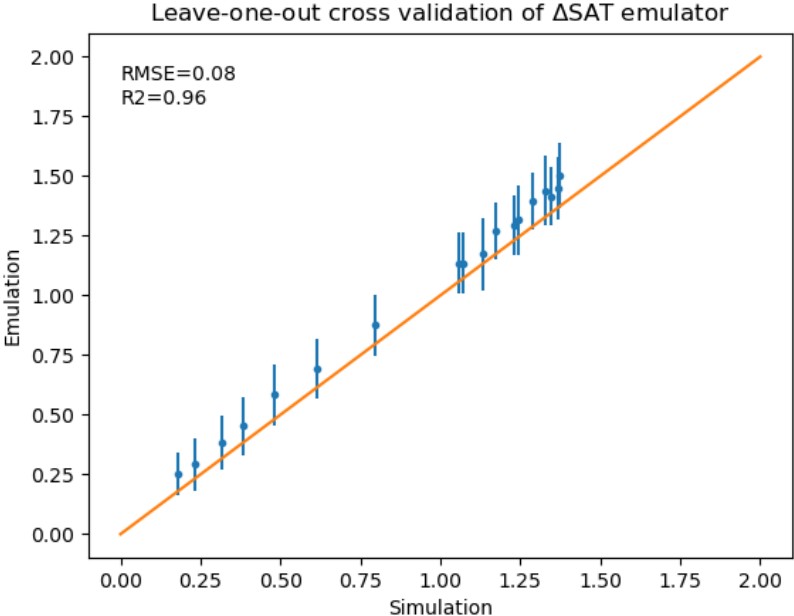

**Figure A2: Emulated value plotted against the simulated value for every 5th timestep (between 2014 and 2100) of a random ensemble member**

We assess the emulator's performance using the root mean square error (RMSE) and the coefficient of determination ($r^2$),

demonstrating that it effectively captures the behavior of the omitted simulation. Even though each time step is emulated independently and the temporal correlation between different timesteps are not known by the emulators, the whole emulated timeseries matches the simulated one well in this case (as indicated by the high $r^2$). A comprehensive summary of all validated timesteps across all simulations is shown in Figure A3.

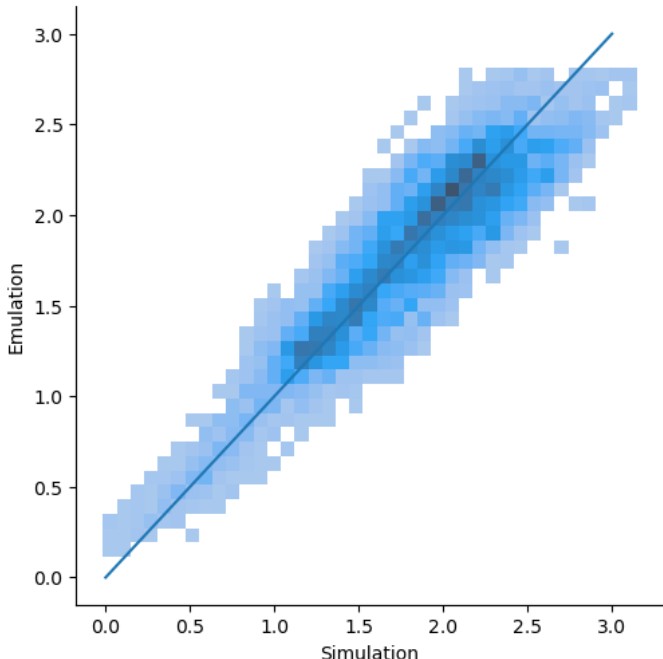

**Figure A3: Distribution of emulated versus simulated points**

While some emulators exhibit lower performance, the vast majority produce emulated values that closely match the simulated ones. For each simulation, the RMSE is calculated for the time series and averaged across the entire ensemble, yielding a mean RMSE of 0.19°C, an error considered reasonable. The average coefficient of determination (r2) is 0.97, indicating also a strong agreement between the emulated and simulated values.


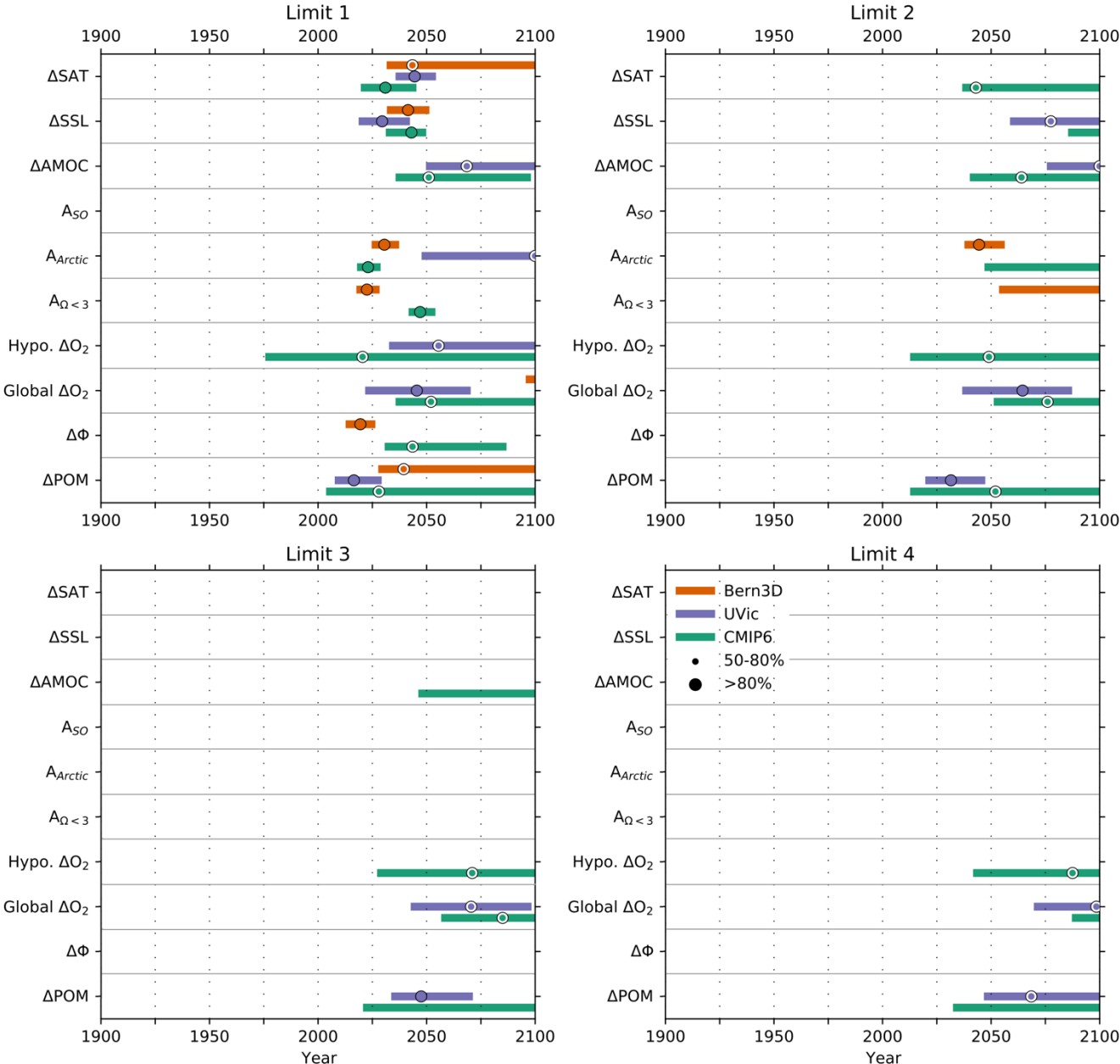

**Figure A4: Same as in Figure 7 but for SSP1-2.6.**

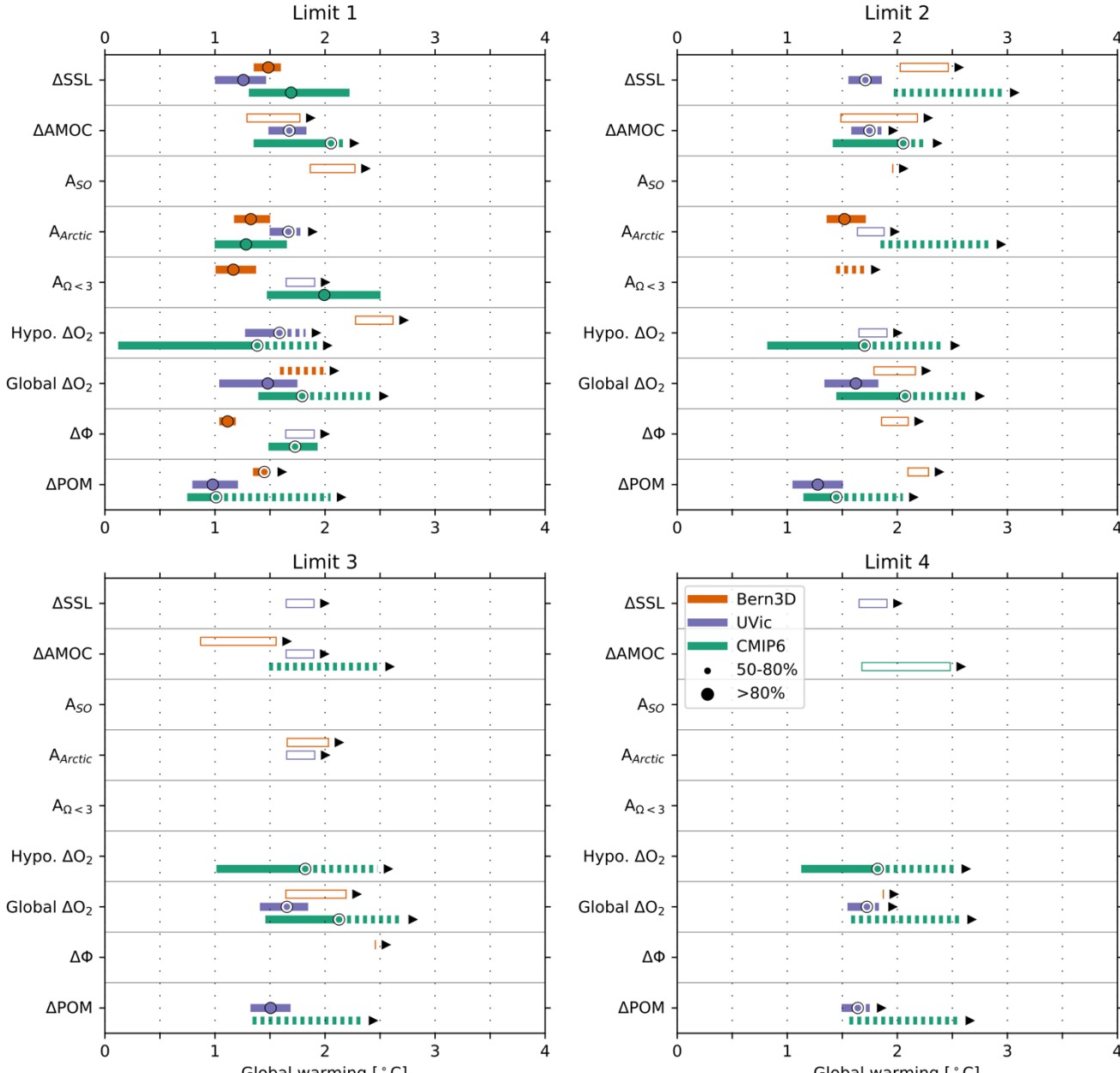

**Figure A5: Same as in Figure 8 but for SSP1-2.6.**

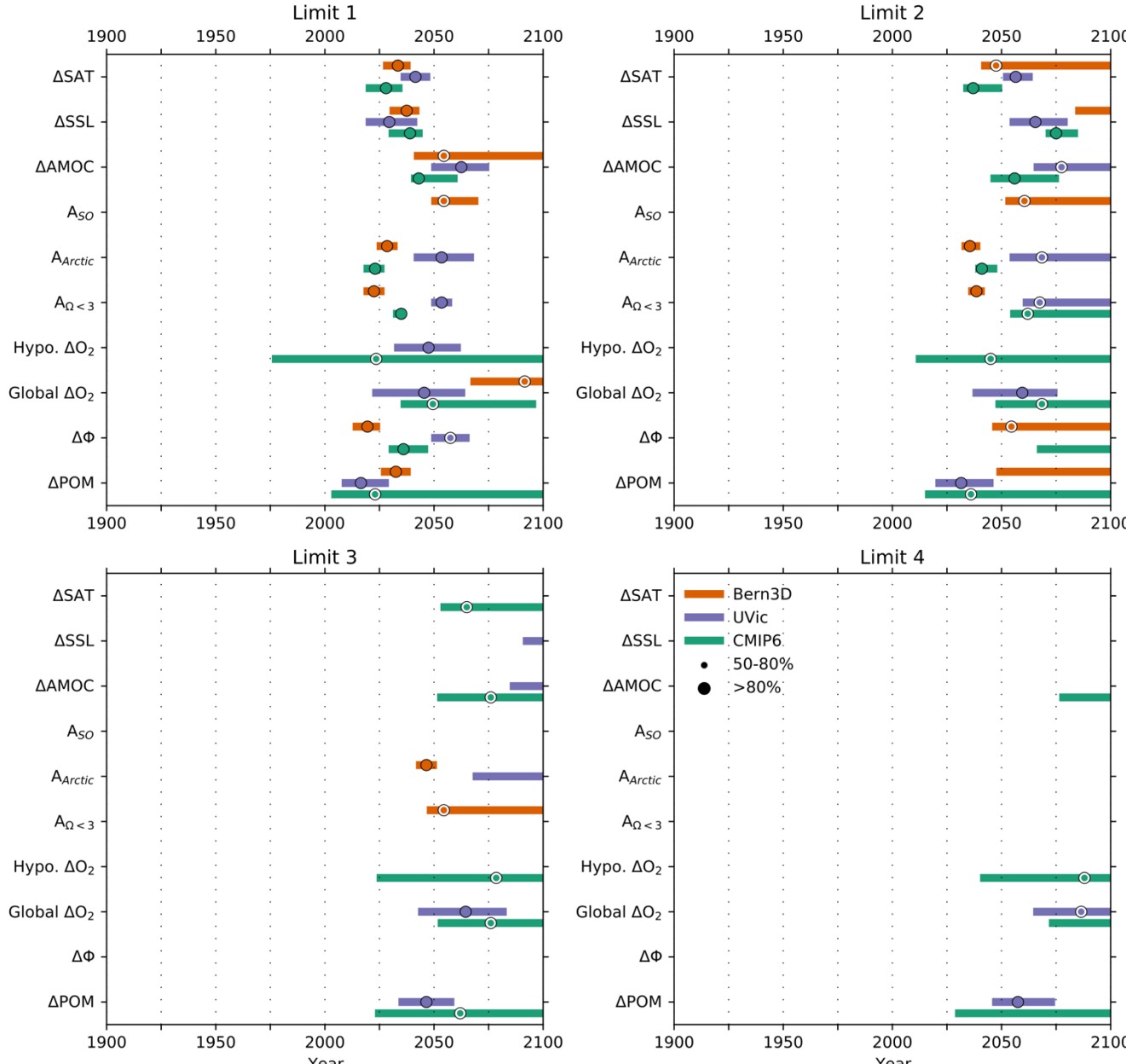


**Figure A6: Same as in Figure 7 but for SSP5-3.4-OS.**

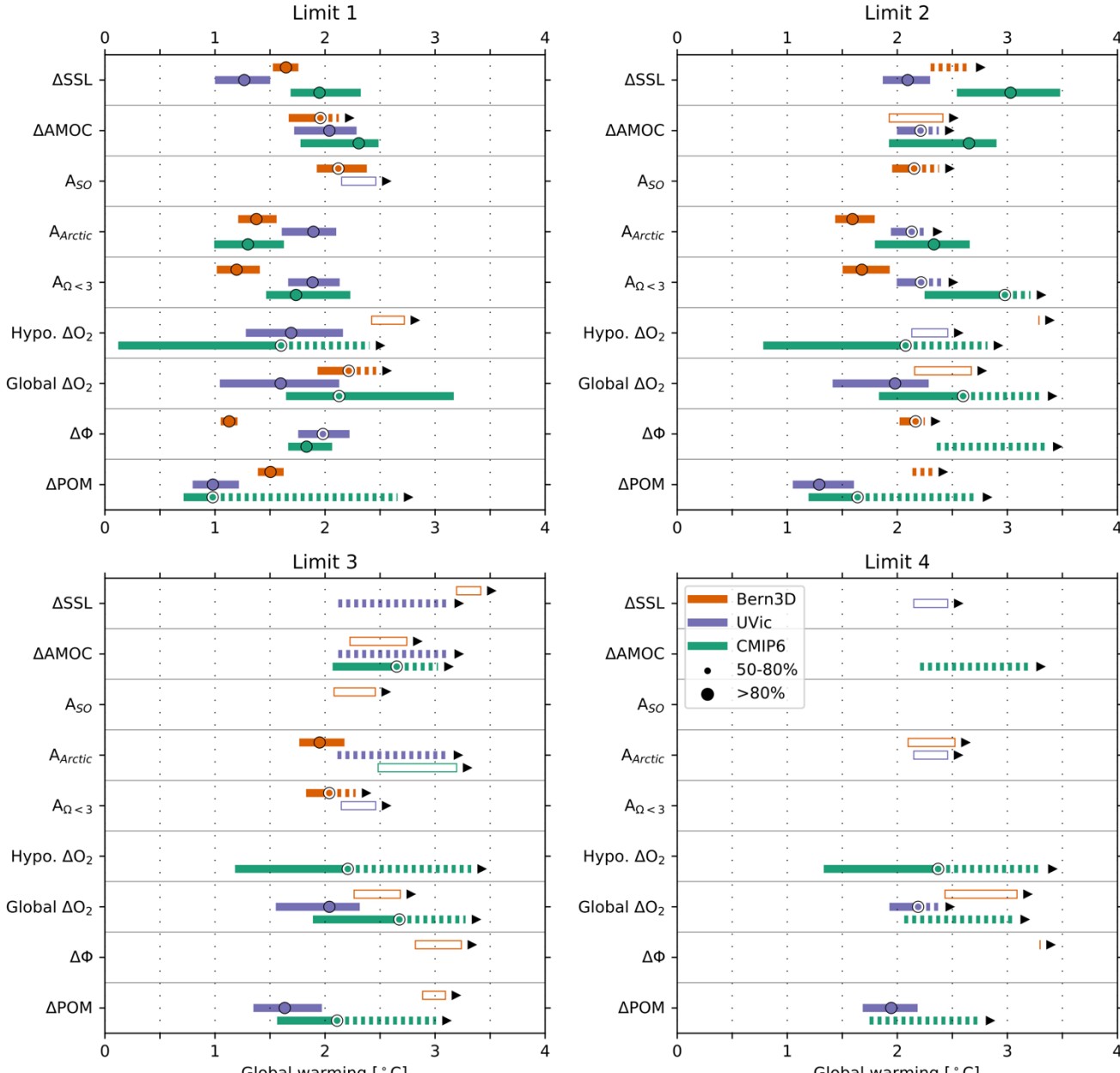

**Figure A7: Same as in Figure 8 but for SSP5-3.4-OS.**

## Code availability

The mocsy 2.0 code is publicly available via https://github.com/jamesorr/mocsy (Orr and Epitalon, 2015).

**Data availability**

CMIP6 outputs are publicly available from the Earth System Grid Federation (ESGF) portals (e.g., https://esgf-data.dkrz.de/). The World Ocean Atlas 2013 (Locarnini et al., 2013; Zweng et al., 2013; https://www.nodc.noaa.gov/OC5/woa13/) and the GLODAPv2 (Lauvset et al., 2016; https://www.nodc.noaa.gov/ocads/oceans/GLODAPv2_2019/) data products are available from the National Oceanographic Data Center portal of the National Oceanic and Atmospheric Administration.

**Author contribution**

The study was led by T.B. who performed the CMIP6 calculations, analysis, and figures. O.T. processed the CMIP6 model data (download, regridding). F.F. provided the CMIP6 metabolic index data. A. J.-T. analysed the Bern-3D data and provided the EMICs figures. G. T. T. led the metrics' and limits' literature review and analysed the UVic data. T. L. F. provided the CMIP6 marine heatwave data. J. N. provided the summary Figs. 5 and 6. All authors were involved in designing the analysis, interpreting the results and writing the manuscript.

**Competing interests**

The authors declare that they have no conflict of interest.

**Disclaimer**

The work reflects only the authors' view; the European Commission and their executive agency are not responsible for any use that may be made of the information the work contains.

**Acknowledgments**

We acknowledge the World Climate Research Programme, which, through its Working Group on Coupled Modelling, coordinated and promoted CMIP. We thank the climate modelling groups for producing and making available their model output, the Earth System Grid Federation (ESGF) for archiving the data and providing access, and the multiple funding agencies who support CMIP and ESGF. All authors received funding from the European Union's Horizon 2020 research and innovation programme under grant agreement No. 820989 (COMFORT). TB and JS received also funding from the European Union's Horizon Europe research and innovation programme under grant agreement No. 869357 (OceanNETs). FJ and AJ acknowledge additional funding by the Swiss National Science Foundation (#200020_200511). TLF and FF received also funding from the European Union's Horizon Europe research and innovation programme under grant agreement No. 101137673 (TipESM). We also thank the IPSL modelling group for the software infrastructure, which facilitated CMIP6

analysis. We thank NORCE Research and the Bjerknes Centre for Climate Research for covering article processing charges. We also thank Pearse Buchanan and one anonymous reviewer for their helpful comments on our manuscript and the associate editor Jack Middelburg for handling our manuscript.

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
