# Peer review of "Mapping the safe operating space of marine ecosystems under contrasting emission pathways"

_EGUsphere, 2024_

## Referee Comment (RC1)

**Bourgeois et al. – Pathways for avoiding ocean biogeochemical damage: Mitigation limits, mitigation options, and projections.**

**Summary**

Bourgeois and colleagues investigate the date when "safe operating limits" are crossed with regard to ocean properties under 3 different climate change scenarios over the next century. They predefine 4 limits for each of 14 properties, from safest (least change) to the least safe (most change) based on their review of the literature. Of their 14 properties, they consider 5 physical properties, 5 chemical properties, and 4 ecosystem properties. They assess time at which limits are crossed using a multi-model ensemble, and in doing so, they are able to provide a a measure of mean ± range in the date, plus a measure of confidence based on how many models agree. High confidence is afforded to those limits that are passed by > 80% of models, and low confidence when < 50% of models show an exceedance of the limit.

The authors present their analysis in 8 Figures and discuss, in simple terms, when exceedances of the limits are passed for each of the 14 properties. They also assess the global warming levels at which point some of the limits are passed, which is a nice addition to their study.

I have some recommendations that I hope the authors can take into account to improve the manuscript.

**Main recommendations:**

1. After reading the paper, I came away with little clarity in what I had learned. That might seem harsh, but what I am asking for is greater focus in the communication of the main results. There is a lot of data, and given that there are 14 properties with 4 limits, a lot of information to condense. This expansive thinking and data analysis is admirable. However, I think it is important to focus in on some key properties, one at a time, or perhaps separate them into groups (physical, chemical, biological) and address when safe limits are passed in a more systematic manner. This way, a biologist can come along to your paper and immediately seek out the "biology" section and get a sense of when some limits might be passed that are relevant to their work.

2. Another suggestion I have is to place your "safe operating limits" within an ecological, socio-economic or geopolitical context. Everything depends on this. You start with this in your definition of them, but I ask that you then circle back to this at the end within a discussion section (currently not existent). So that I try to be constructive with this suggestion, I will try an example: The collapse of the AMOC or at least its weakening would have socio-economic and geopolitical ramifications, as well as ecological as AMOC transports a lot of heat and nutrients to the subpolar North Atlantic. Passing of AMOC limits might interact with the passing of NPP and biomass limits in the North Atlantic, affecting fisheries as well as temperature and thereby affecting food supply and energy demands of the region. It would also cause heating in the Southern Hemisphere that may cause more extremes in places like Australia, Southern Africa, ect. These sorts of considerations are essential to make your results concrete and tangible for the reader. Everything depends on context setting. Without

this, I finish the paper and think very little of it, not knowing what I've learned and what happens if we cross limit 2. And, it will also help improve your citation of the literature, which at the moment seems inadequate to me when finishing the paper.

3. I also feel that the presentation of the results needs some extra thinking. The statement starting on line 315 that "For the less ambitious mitigation limits, exceedance time estimates generally move towards later times and higher warming levels", doesn't seem to be the case. Looking at Figures 1 and 2 shows that for most properties, the exceedance time is generally *earlier* for the SSP1-2.6 scenario than the SSP5-8.5 scenario. This is the case specifically for:
   - MHW duration
   - Metabolic index
   - Change in biomass
   - Change in AMOC
   - Change in SSL
   - Change in POM flux
   - Global change in dissolved oxygen

   From my reading of Figures 1 and 2 it seems to be rarely the case that "estimates generally move towards later times". Only global change in oxygen at limit 3, and the area of $\Omega_A > 3$ at limit 1 seem to satisfy this statement. I understand and agree with the fact that less models under SSP1-2.6 are exceeding the limits, so the dates that are presented in Figures 1 and 2 are those built from a smaller subset of models that are almost certainly more sensitive. Meanwhile, later exceedances under SSP5-8.5 are the cause of including the less sensitive models. This makes me wonder if this is the right way to present the data… I think the authors should go away and reconsider how to present this information so that they more appropriately present the times of exceedance, because without a careful reading of the paper as it currently stands, the reader will come away confused about the fact that SSP1-2.6 appears more extreme in its effects on the ocean than SSP5-8.5. Perhaps the models themselves need to be compared individually, or grouped into sensitive and un-sensitive models?

4. Another criticism I have with the paper is the use of an 1850-1900 baseline to assess marine heatwave duration. As the authors report, this results in a near-permanent heatwave by the end of the 21st century under the high-emissions scenario. But, ecologically, this is not so interesting. What is more ecologically interesting is to move the baseline incrementally at a rate that captures some degree of physiological adaptation or evolution. For microscopic organisms, this rate might be fast, while for mammals and other top predators, this baseline might be very slow, if not static like the authors consider here. With a rapidly shifting baseline, the authors would be able to comment on the prevalence of anomalous heating events that marine ecosystems, given an ability to adapt, would still not be physiologically prepared for. The authors could assume two extreme cases: they keep their static baseline of 1850-1900 to reflect on the effects to top predators where adaptation is slow, and consider a moving baseline that is always positioned a few years earlier (e.g., 20-year climatology of 1991-2010 when assessing extremes in 2011.) to reflect a rapidly adapting

group of microscopic organisms (e.g., Jin and Agustí, 2018). This also opens the door to extreme cooling phases in the overshoot scenario, which would offer a unique perspective.

5. I also think the description of the EMICs was lost on me. What is the importance of these models and why did you use them? Why do we need to know the specifics about their ensembles and the optimisation of the UVic? Why was this optimisation important for your work? What is the skill of the emulator? What does it tell us if the EMICs and CMIP6 models disagree? I think you have to demonstrate that the use of the EMICs was qualitatively essential to your conclusions, and I do not understand that from the writing as it currently stands.

**Specific comments (note that I have held back on specifics until the main comments are addressed):**

- Figures 1 & 2: Can you put the x-axis labels (years or global warming level) on the top of the panels as well? It is hard to see the year at which MHW limits are exceeded, for instance.
- Figures 1 & 2: Can you arrange your y-axis categories in order of introduction in your methods? First do the 5 physical properties, then the 5 chemical, then the 4 biological?
- Figure 6 legend needs to say why it doesn't include SAT, even though it's obvious. Just to help the reader out a bit.
- I think it would be better to spell out the properties, rather than use the short hand symbols. For example, use Southern Ocean $\Omega_A < 1$ rather than $A_{SO}$.
- Line 293: The findings of which were what?
- Line 294: Why were substantial uncertainties in O2 found?
- Line 315: What does "less ambitious mitigation limits" mean?

Thank you for considering my input to your research,

Pearse J. Buchanan.

---

## Author Response (AR1)

**Author's response to referee comments on egusphere-2024-2768**

We thank Dr. Pearse Buchanan and Anonymous Referee #2 for their thoughtful comments and suggestions. We have addressed all points raised in the reviews as detailed in our point-by-point response below. Please note additional remarks independent from the referee comments, placed before the References section of this document.

The referee comments are shown in italic grey font while our point-by-point replies including the relevant changes made in the manuscript are shown in black font. Due to the major revisions applied to the revised manuscript, we did not copy-paste here all the changes included in the revised manuscript and we refer the reviewers to the author's track-changes version of the manuscript.

**RC1: 'Comment on egusphere-2024-2768', Pearse Buchanan, 05 Nov 2024**

*Main recommendations:*

1. *After reading the paper, I came away with little clarity in what I had learned. That might seem harsh, but what I am asking for is greater focus in the communication of the main results. There is a lot of data, and given that there are 14 properties with 4 limits, a lot of information to condense. This expansive thinking and data analysis is admirable. However, I think it is important to focus in on some key properties, one at a time, or perhaps separate them into groups (physical, chemical, biological) and address when safe limits are passed in a more systematic manner. This way, a biologist can come along to your paper and immediately seek out the "biology" section and get a sense of when some limits might be passed that are relevant to their work.*

We improved the clarity of the manuscript by expanding the implications of our study in a renamed section "4 Implications and Conclusion". This section includes subsections focusing respectively on physical changes, ocean acidification, and biogeochemical/biological changes.

2. *Another suggestion I have is to place your "safe operating limits" within an ecological, socioeconomic or geopolitical context. Everything depends on this. You start with this in your definition of them, but I ask that you then circle back to this at the end within a discussion section (currently not existent). So that I try to be constructive with this suggestion, I will try an example: The collapse of the AMOC or at least its weakening would have socio-economic and geopolitical ramifications, as well as ecological as AMOC transports a lot of heat and nutrients to the subpolar North Atlantic. Passing of AMOC limits might interact with the passing of NPP and biomass limits in the North Atlantic, affecting fisheries as well as temperature and thereby affecting food supply and energy demands of the region. It would also cause heating in the Southern Hemisphere that may cause more extremes in places like Australia, Southern Africa, ect. These sorts of considerations are essential to make your results concrete and tangible for the reader. Everything depends on context setting. Without this, I finish the paper and think very little of it, not knowing what I've learned and what happens if we cross limit 2. And, it will also help improve your citation of the literature, which at the moment seems inadequate to me when finishing the paper.*

We agree that translating the meaning of our results to the larger ecological, socioeconomical and geopolitical systems is appealing for policy relevance, and we appreciate that the referee provides such an example. However, as mentioned in the Methods, many metrics suffer from a lack of knowledge regarding the assessment of actual impacts that an exceedance would have on the Earth system or ecosystem functioning. This is particularly true for economic and geopolitical impacts. The discussion of such aspects would be largely speculative and not sufficiently underpinned by the model results analysed in this study.

Nevertheless, we decided to rename our Conclusion section "Implications and conclusions" and expanded it thoroughly to better emphasize the implications of our results.

> 3. *I also feel that the presentation of the results needs some extra thinking. The statement starting on line 315 that "For the less ambitious mitigation limits, exceedance time estimates generally move towards later times and higher warming levels", doesn't seem to be the case. Looking at Figures 1 and 2 shows that for most properties, the exceedance time is generally earlier for the SSP1-2.6 scenario than the SSP5-8.5 scenario. This is the case specifically for:*
>
> - *MHW duration*
> - *Metabolic index*
> - *Change in biomass*
> - *Change in AMOC*
> - *Change in SSL*
> - *Change in POM flux*
> - *Global change in dissolved oxygen*
>
> *From my reading of Figures 1 and 2 it seems to be rarely the case that "estimates generally move towards later times". Only global change in oxygen at limit 3, and the area of WA > 3 at limit 1 seem to satisfy this statement. I understand and agree with the fact that less models under SSP1-2.6 are exceeding the limits, so the dates that are presented in Figures 1 and 2 are those built from a smaller subset of models that are almost certainly more sensitive. Meanwhile, later exceedances under SSP5-8.5 are the cause of including the less sensitive models. This makes me wonder if this is the right way to present the data... I think the authors should go away and reconsider how to present this information so that they more appropriately present the times of exceedance, because without a careful reading of the paper as it currently stands, the reader will come away confused about the fact that SSP12.6 appears more extreme in its effects on the ocean than SSP5-8.5. Perhaps the models themselves need to be compared individually, or grouped into sensitive and un-sensitive models?*

We understand and agree that some aspects of our results are confusing.

As correctly stated by the referee, an issue of our approach was that, in the case of a partially exceeding ensemble for a given limit of a given metric, the exceedance distribution is dominated by high-sensitivity models to this metric, sometimes leading to the counter-intuitive earlier exceedances for lower-emission scenarios. To avoid this, we changed changing

the exceedance definition used in our study by attributing the year 2100 to models not exceeding the limit. When not exceeding the limit, these models are also attributed the highest warming level reached by the model under the given scenario. By using this approach, we include in the exceedance distribution the information from models not exceeding a limit. This approach moves the exceedance distribution toward later time and higher warming levels but remains conservative (i.e., early/lower bound of ensemble uncertainty) because models with 2100 as the default exceedance year and with the highest reached warming level as the default exceeding global warming level are likely to (1) exceed the limit later than 2100, (2) exceed the limit in a warmer global warming level than the maximum warming level reached in a given scenario, or (3) never exceed the limit. This approach removes most of the counter-intuitive results. However, even with a 20-year moving averaging, metrics with large interannual to decadal variability such as the AMOC strength can still show exceedance distribution not entirely consistent with the different radiative forcing levels applied in the emissions scenarios. In such metrics, difference between scenarios in exceedance years of around 10-15 years still occur, but are not significantly different (i.e., they can be considered as occurring simultaneously). In addition, we decided to not show any data when none of the models exceed a given limit.

New symbols/bars have been introduced for visualizing the results over the temperature dimension (Figure 2) to emphasize the resulting uncertainty and conservative aspects of the exceedances (i.e., warmer levels likely needed).

[Figure]

**Revised Figure 1** using the new exceedance definition suggested in our reply (i.e. assigning the exceedance year 2100 to models not exceeding a limit until 2100). The figure includes suggestions from the referees: (1) replace Subsurf. $\Delta O_2$ by Hypo. $\Delta O_2$, (2) X axis labels duplicated on top of each panel, (3) addition of a metric related to marine heatwave computed from a moving baseline (MHW$_{mov}$, see next comment), and (4) reordering of the metrics following the order in Table 1. MHW$_{fix}$ is now expressed as an anomaly relative to the period 1850-1900. The figure is simplified to ease the interpretation by removing the visualization of outliers / minimums / maximums (crosses and whiskers) and only includes bars showing the $25^{th} - 75^{th}$ percentiles, and dots for the medians where the dot's size reflect the percentage of exceeding models (abbreviated exceeding rate). Limits for $\Delta$SSL have also been updated (see further down for details). Finally, the CNRM-ESM2-1 model has been excluded for the $\Delta$Biomass metric (see further down for details).

[Figure]

**Revised Figure 2** using the new exceedance definition suggested in our reply and including modifications stated in Figure 1. The X axis has been also extended from 4°C to 6°C following Referee #2's suggestion. Very low probability cases with <25% exceeding rate are displayed as empty rectangles only for the reader's information, using the $25^{th}$ — $75^{th}$ percentile range as in Figure 1, but without showing median symbols. Low probability cases in the 25—50% exceeding rate range are represented by continuous dashed bars covering the $25^{th}$ — $75^{th}$ percentile range, without median symbols either. Indeed, medians are meaningless for cases with less than 50% exceeding rate. Black triangles interpreted as arrows pointing toward warmer levels shows explicitly that warmer levels are likely needed to exceed the given limit. Such arrows are added to all cases in the 50—100% exceeding rate (100% exceeding rate excluded). Medium probability cases with 50—75% includes plain bars over the $25^{th}$ — $50^{th}$ percentile range, median symbols, and dashed bars over the $50^{th}$ — $75^{th}$ percentile range to highlight the uncertainty of the $50^{th}$ — $75^{th}$ percentile range. Finally, high probability cases with exceeding rates >75% follow Figure 1's legend (plain bars over $25^{th}$ — $75^{th}$ percentile range with median symbol).

[Figure]

**Revised Figure 3**

[Figure]

**Revised Figure 4**

[Figure]

**SSP5-8.5**

**Revised Figure 7**

[Figure]

**SSP5-8.5**

**Revised Figure 8**

Due to the modification of the results, we changed extensively the text describing and interpreting these figures accordingly in the revised manuscript. All the figures have been updated accordingly.

4. *Another criticism I have with the paper is the use of an 1850-1900 baseline to assess marine heatwave duration. As the authors report, this results in a near-permanent heatwave by the end of the 21st century under the high-emissions scenario. But, ecologically, this is not so interesting. What is more ecologically interesting is to move the baseline incrementally at a rate that captures some degree of physiological adaptation or evolution. For microscopic organisms, this rate might be fast, while for mammals and other top predators, this baseline might be very slow, if not static like the authors consider here. With a rapidly shifting baseline, the authors would be able to comment on the prevalence of anomalous heating events that marine ecosystems, given an ability to adapt, would still not be physiologically prepared for. The authors*

*could assume two extreme cases: they keep their static baseline of 1850-1900 to reflect on the effects to top predators where adaptation is slow, and consider a moving baseline that is always positioned a few years earlier (e.g., 20-year climatology of 1991-2010 when assessing extremes in 2011.) to reflect a rapidly adapting group of microscopic organisms (e.g., Jin and Agustí, 2018). This also opens the door to extreme cooling phases in the overshoot scenario, which would offer a unique perspective.*

As the referee correctly points out, using a fixed preindustrial baseline, marine heatwave conditions are expected to become increasingly common as mean ocean surface temperatures rise. This could lead to "permanent" marine heatwave states in regions experiencing a high level of warming or small variability (Frölicher et al., 2018; Oliver et al., 2019). Nevertheless, these changes are ecologically important, as they may reflect the growing risks such events pose to marine organisms, especially those with slow adaptation rates, such as warm-water corals (Smith et al., 2025).

In response to the referee's suggestion, we have introduced an additional marine heatwave metric: marine heatwaves defined under shifting-mean baseline, $MHW_{shift}$, where the baseline is adjusted according to the forced mean trends in sea surface temperature (Figure 1 and 2). The forced mean trend was identified using a smoothing "Enting" spline (Enting, 1987) with a 80-yr cut off period. To account for model-dependent internal variability, we use the anomaly of $MHW_{mov}$ relative to the period 1850-1900. To homogenize with the former MHW metric (renamed $MHW_{fix}$), we update the definition of this metric by also computing its anomaly relative to 1850-1900.

We changed the text in the Method section to:
"

We consider marine heatwaves due to their substantial global and regional impacts on marine ecosystems (Capotondi et al., 2024; Frölicher and Laufkötter, 2018; Smith et al., 2021). We use two definitions of marine heatwaves based on different baselines (Burger et al., 2022; Smith et al., 2025).

First, we define a marine heatwave day as the local daily mean sea surface temperature (CMIP6 variable tos) exceeding the 90th percentile relative to a fixed seasonally varying 1850-1900 baseline (metric abbreviated $MHW_{fix}$). In this case, changes in marine heatwaves are driven by both long-term surface ocean warming trends and changes in anthropogenically-forced internal variability. The fixed baseline may be particularly relevant for assessing the risk marine heatwaves pose to organisms with slow adaptation rates.

Second, we define a marine heatwave day relative to a shifting-mean baseline ($MHW_{shift}$), where the 1850-1900 percentile thresholds are adjusted according to the forced mean trend in sea surface temperature (SST). The forced trend is identified using a smoothing "Enting" spline (Enting, 1987) with a 80-yr cut off period. In the shifting-mean approach, changes in marine heatwave duration are primarily driven by changes in anthropogenically-forced internal variability, while the long-term warming trends is already accounted for in the baseline (Burger et al., 2022; Deser et al., 2024). The choice between a fixed or shifting baseline depends on the specific application. For example, the shifting-mean case may better capture the risks posed to organisms that can adapt to long-term warming trends.

For both definitions, we (1) calculate the global annual mean duration of marine heatwaves, and (2) deduce the anomaly relative to the 1850-1900 period to normalize model-dependent internal variability. Given the lack of strong observational constraints on a global marine heatwave exceedance metric, we distribute uniformly the mitigation limit values of $MHW_{fix}$ over the year as 90, 180, 270, 360 (i.e., permanent heatwave) days while we distribute the limits of $MHW_{shift}$ over the range of projected values under the scenarios used in this study with 4, 6, 8, and 10 days."

We enriched the discussion related to MHW in the Implications and conclusion section.

> 5. *I also think the description of the EMICs was lost on me. What is the importance of these models and why did you use them? Why do we need to know the specifics about their ensembles and the optimisation of the UVic? Why was this optimisation important for your work? What is the skill of the emulator? What does it tell us if the EMICs and CMIP6 models disagree? I think you have to demonstrate that the use of the EMICs was qualitatively essential to your conclusions, and I do not understand that from the writing as it currently stands.*

We agree that the justifications on the use of EMICs could have been made clearer. These tools are important in the scope of our study because of their small computational resource consumption compared to ESMs, permitting to conduct sensitivity analysis using large perturbed-parameter or initial conditions ensembles for uncertainty quantification with simulations over longer time scales (1000s years), particularly suited for applications such as CDR scenarios (Jeltsch-Thömmes et al., 2024), but also for approaches similar to those presented in this study (Steinacher et al., 2013). Yet, EMICs are rarely compared to ESMs, and we believe our study fills an important gap in this respect.
Regarding the importance of the optimisation for EMICs, this method is used to quantify the uncertainties arising from uncertainties in model parameters EMIC's perturbed-parameter ensembles (PPE) allow us to analyse the uncertainty within a single model due to uncertain parameters, which is rarely quantified in ESMs. Ideally, we would like to have multi-model PPEs but that is currently not available for comprehensive models like CMIP6 ESMs so we used UVic and Bern3D.

Regarding to the skill of the UVic ESCM's emulator, we added the following information about the emulator validation in the appendix:
"
To evaluate the performance of the emulator, we employ leave-one-out cross-validation to assess the quality of our Gaussian process emulators. This validation is conducted using the surface air temperature anomaly ($\Delta$SAT) relative to the preindustrial period, spanning the years 2015 to 2100. For our ensemble of 325 simulations, we iteratively exclude one simulation and construct an emulator of $\Delta$SAT using the remaining 324 simulations. The emulator is then used to predict the $\Delta$SAT of the omitted simulation. This process is repeated for each simulation in the ensemble, ensuring that every simulation is excluded once. Additionally, validation is performed at every 5th timestep in the time series.

Figure A1 presents an example of the emulated versus simulated ΔSAT values at every 5-time steps for a randomly selected ensemble member. The error bars represent the two standard deviations of the predicted mean estimated by the emulator. For a well-calibrated emulator, approximately 95% of the true or simulated values should fall within 2 standard deviations of the emulated values.

[Figure]

Figure A1: Emulated value plotted against the simulated value for every 5th timestep (between 2014 and 2100) of a random ensemble member.

We assess the emulator's performance using the root mean square error (RMSE) and the coefficient of determination ($r^2$), demonstrating that it effectively captures the behavior of the omitted simulation. Even though each time step is emulated independently and the temporal correlation between different timesteps are not known by the emulators, the whole emulated timeseries matches the simulated one well in this case (as indicated by the high $r^2$). A comprehensive summary of all validated timesteps across all simulations is shown in Figure A2.

[Figure]

Figure A2: Distribution of emulated versus simulated points

While some emulators exhibit lower performance, the vast majority produce emulated values that closely match the simulated ones. For each simulation, the RMSE is calculated for the time series and averaged across the entire ensemble, yielding a mean RMSE of 0.19°C, an error considered reasonable. The average coefficient of determination ($r^2$) is 0.97, indicating also a strong agreement between the emulated and simulated values."

Regarding the meaning of EMIC/CMIP6 disagreement, we emphasized in the introduction section the need for modelling studies to use diverse modelling tools instead of single-type models (only ESM(s), or only EMIC(s)). Both are tools that can be used to investigate specific aspects of a question (e.g., regional/high-resolution, parameter or initial conditions uncertainty, variability, extreme events). It is interesting to know whether these models agree and if not, where they disagree. As of why, this question is out of the scope of this study.

We enriched the manuscript with all the above EMIC-related information.

Regarding to the inequal description between the Bern3D and UVic models, the lengthier method section related to UVic compared to Bern3D is explained by the fact that Bern3D methodology is already published (Jeltsch-Thömmes et al., 2024) contrary to UVic's. We reduced the UVic-related section to homogenize with the Bern3D section and added the removed information to the appendix for the reader's information.

*Specific comments (note that I have held back on specifics until the main comments are addressed):*

- *Figures 1 & 2: Can you put the x-axis labels (years or global warming level) on the top of the panels as well? It is hard to see the year at which MHW limits are exceeded, for instance.*

Yes, thank you for the suggestion. This has been done, see revised Figures above.

- *Figures 1 & 2: Can you arrange your y-axis categories in order of introduction in your methods? First do the 5 physical properties, then the 5 chemical, then the 4 biological?*

Yes, thank you for the suggestion. This has been done, see revised Figures above.

- *Figure 6 legend needs to say why it doesn't include SAT, even though it's obvious. Just to help the reader out a bit.*

Yes, thank you for the suggestion. We did so to avoid misunderstandings.

- *I think it would be better to spell out the properties, rather than use the short hand symbols.*
  *For example, use Southern Ocean $\Omega_A < 1$ rather than $A_{SO}$.*

Thank you for the suggestion. We did so to increase the readability in the text.

- *Line 293: The findings of which were what?*

Indeed, we explicitly state now the findings of Tittensor et al. (2021) that plankton biomass is a more robust and relevant impact metric than NPP with respect to impacts of climate change on marine ecosystems.

- *Line 294: Why were substantial uncertainties in O2 found?*

The substantial uncertainty found in Cocco et al. (2013) can be explained by the uncertain balance between $O_2$ supply from physical mixing and advection, and $O_2$ consumption from remineralization of organic matter. Uncertainties remains also on how these processes would respond to future rising $CO_2$. Regarding subsurface $O_2$ projections, Frölicher et al. (2016) identified model structure and parametrization as the second source of projection uncertainty after the scenario uncertainty already accounted in our study. These aspects are now added to the revised manuscript.

- *Line 315: What does "less ambitious mitigation limits" mean?*

The 4 mitigation limits can be also interpreted as requiring decreasing levels of ambition regarding emission reductions. The first level being the most ambitious, characterized by minimal change from the pre-industrial baseline and minimal impacts on the ocean-climate-society nexus. Thus, less ambitious mitigation limits refer to higher levels, such as the third and/or fourth level. This aspect is already defined in the first paragraph of the Methods section 2.1, so we suggest to not repeat it here.

**RC2: 'Comment on egusphere-2024-2768', Anonymous Referee #2, 15 Nov 2024**

*Bourgeois et al. identified 14 different climate sensitive metrics related to physics and ocean biogeochemistry, and set criteria for 4 levels of limits based on existing literature, then analyzed a suite of ESMs and ESIMs future projections to estimate whether and when these limits are reached as the climate warms, and how this differs in response to different emission scenarios. Bias corrections are applied where necessary, and uncertainties are carefully considered. Overall, I found this analysis useful for bridging CMIP ensemble output variables and more societally relevant metrics, tipping points, and probability of occurrences, and a good contribution to the current understanding of CMIP datasets as well as future CMIP type of efforts.*

Thank you for the positive feedback.

*I have a few quibbles and requests listed below:*

1. *The authors chose the 4 levels of limits based on literature, which is reasonable. However, some of the literature is based on observations or single events, thus it might be challenging to use CMIP models, which are mostly at coarse and intermediate resolution, to capture the observed ranges. This results that even Level 1 limit is never reached with high uncertainty for a number of analyzed metrics. I suggest that the authors use literature as a guidance, but also consider the model ensemble's capability of capturing those ranges and changes. One thing that could be looked at is the historical period of the CMIP simulations between 1950-2015. If literature suggests that metric A has changed by xx% over this period yet the model ensemble can only capture about yy% (e.g., xx>>yy), then the authors may want to consider scaling the limits accordingly. This may help minimize the loss of signals solely due to choices made on an ad hoc basis. The authors indeed applied some levels of bias corrections for a few metrics but it was done in a systematic way.*

Referee #2 suggests checking model-observations discrepancies over the historical period and, if so, scale the limits accordingly (e.g., decrease the limits of a given metric if the model ensemble tends to underestimate its historical change). We agree that scaling the limits would be ideal but many of our metrics do not have suitable observation-based trend analogues and sufficiently long timeseries describing historical trends. On the other hand, many metrics are either already corrected or the CMIP6 model ensemble spread has been already evaluated as being within the uncertainty range of observation-based trends.

Here is a quick review of the metrics regarding to this suggestion:
- ΔSAT: The observed global warming trend is within the CMIP6 ensemble spread.
- MHW: CMIP6 models overestimate the trends in marine heatwave durations according to Plecha and Soares (2020). However, both observation- and ESM-based present-day climatologies of MHW events generally overestimates their duration, mainly due to the coarse resolution used in models, or the interpolation methods used in observation-based data products. This makes a proper comparison difficult.
- ΔSSL: Already corrected from potential model drift using the piControl experiment. However, we acknowledge that, here, our limits are too high and do not acknowledge the already increasing frequency of coastal floods induced by sea level rise (Hague et

al., 2023; Oppenheimer et al., 2019). As mentioned in the manuscript, the SSL rise is estimated today at around 0.08 m. We decreased the limits of SSL rise by 0.1 m, leading to limits of 0.1, 0.2, 0.3, and 0.4 m.

- Arctic SIE: The trend in observed Arctic sea ice extent is within the CMIP6 ensemble spread (Notz and SIMIP community, 2020).
- AMOC: The time series of the RAPID 26°N transect is too short to deduce a trend (Lobelle et al., 2020).
- $\Omega_{arag}$: Bias correction using Terhaar et al. (2020) methodology.
- Subsurface $O_2$: No correction applied because the CMIP6 climatology and trend are within the observation range (Bindoff et al., 2019; Séférian et al., 2020).
- Global $O_2$: Already corrected from potential model drift using the piControl experiment.
- Metabolic index: No observation-based trends found.
- NPP: No global observation-based trends found.
- Carbon export: No observation-based trends found.
- Metabolic index: No global observation-based trends found.

2. *Line 97: "we spread evenly …" I am not clear what this means. Please clarify.*

We replaced "spread evenly" by "distribute uniformly".

3. *Line 103: "… 40% of the total sea-level rise of 0.2 m today" - is the approximated SSL rise 0.2 m or 0.08 m, then? The chosen levels of limits are 0.2, 0.3, 0.4, and 0.5, so I am assuming SSL rise is about 0.2 m based on literature. Please clarify the original sentence.*

This comment refers to the following sentence: "The SSL rise is approximated to be 40 % of the total sea-level rise of 0.2 m today". To clarify, we replaced this sentence by the following: "The SSL rise is estimated to account for 40 % of the total sea-level rise of 0.2 m today, i.e., the estimated SSL rise is 0.08 m."

Our chosen limits of 0.2, 0.3, 0.4, and 0.5 m of SSL rise was reflecting a first assessment of the current state of the literature in the submitted manuscript. However, in the former reply to referee comments, we noticed that the first limit proposed in the submitted manuscript (0.2 m) is too far from the current SSL rise (0.08 m).

As mentioned earlier, we reduced the limits related to SSL rise by 0.1 m in the revised manuscript, i.e., 0.1, 0.2, 0.3, and 0.4 m, to acknowledge recent literature on the current impact of anthropogenically-induced SSL rise on coastal flooding (Hague et al., 2023).

4. *Line 117 through 125: despite shown in Table 1, the AMOC metric and its four limits are not clearly described as for other metrics.*

We agree that we did not elaborate enough on the choices regarding the definition of the AMOC metric and its four limits. We use the same definition used in Weijer et al. (2020) to compute the strength of the AMOC and we add justifications of the lack of knowledge on

AMOC variability to choose limits covering the typical range of model responses (Weaver et al., 2012; Weijer et al., 2020).

Thus, we replaced the two sentences line 123-125 by the following:
"We compute the strength of the AMOC as the vertical maximum of the stream function at 26°N following Weijer et al. (2020). As for the limits, despite a growing body of literature on the historical and projected evolution of the AMOC, we still lack sufficiently long observation-based time series, knowledge, and scientific consensus to understand if the AMOC is already experiencing a decline exceeding natural variability, and if such decline is attributed to anthropogenic forcing (Jackson et al., 2022; Latif et al., 2022; Lobelle et al., 2020; Terhaar et al., 2025). Due to the absence of more robust knowledge, we choose four limits at 20, 25, 30, and 40% decline to cover the typical range of model responses (Weaver et al., 2012; Weijer et al., 2020)."

> 5. *Line 129 "ocean acidification could lead to undersaturation and dissolution of calcium ..." This is not accurate. The majority of the ocean interior has an omega value under 1 so OA actually leads to undersaturation and dissolution of calcium carbonate at shallower depths.*

We agree that most of the ocean interior has $\Omega$ values < 1, but the sentence line 129-130 refers to "parts of the surface ocean", similar to "at shallower depth" as suggested by the referee. We left the sentence as is.

> 6. *The "subsurface delta O2" metric and its abbreviation are confusing as they are not consistent with the actual definition of this metric. Suggest to use "hypo(xic) delta O2" which usually refers to the volume change in the relevant research field.*

We agree with this suggestion, and we used "hypoxic $\Delta O_2$", and "Hypo. $\Delta O_2$" instead of "subsurface $\Delta O_2$" or "Subsurf. $\Delta O_2$".

> 7. *Line 175: the calculation of metabolic index needs a bit more clarification. How is it averaged across the 61 species to get a single number? Does biomass distribution matter when averaging?*

We use a metabolic index $\Phi$ calculated as in Fröb et al. (2024) using their median ecophysiotype of 61 species described in Deutsch et al. (2020). Fröb et al. (2024) did not consider the biomass distribution in their statistics. We refer the reader to Fröb et al. (2024) for more details on the methodology but we clarified the sentence line 175 and replace it by: "$\Phi$ has been calculated following Fröb et al. (2024) using the median ecophysiotype of 61 species described in Deutsch et al. (2020), without considering biomass distribution."

> 8. *Line 200-204: all bias corrections should be described in the same place, including aragonite saturation state.*

We agree, and we moved the sentences in lines 144-149 related to bias correction of $\Omega_{arag}$ to the paragraph describing other bias corrections (lines 200-204).

*9. The two ESIMS are described in great detail while the ESMs are not. I suggest the author at least list spatial resolution of the native model grids (even though outputs are regridded) and screamline the description of the two ESIMs.*

We initially agreed to add the ESMs' spatial resolutions in Table 2 but considering that all ESMs ocean component use a nominal 1° resolution, we added the following to section 2.2 describing the CMIP6 ensemble:

"All ESMs use ocean components with a nominal horizontal resolution of about 1° with grid refinements of up to about 1/3° both poleward and at the equator."

The description of the EMICs have now been streamlined, as mentioned in our reply to referee #1.

*10. I think Fig 1 and Fig 2 can be easily combined by adding a second x-axis. I also think it is okay and even preferred to have the warming axis on a different range for each limit level.*

We do not fully understand the Referee's suggestion. If the suggestion refers to adding a global warming axis next to the time axis in Figure 1, this is not possible because of the different global warming pathways of the respective scenarios relative to time. Furthermore, these Figures includes already a lot of information, so we think that splitting it in two Figures improves the readability.

Regarding to changing the range of the warming axis per limit level in Fig. 2, we have now increased the upper bound from 4°C to 6°C in all panels to visualize exceedances with warming level above 4°C. We prefer keeping the same X axis bounds in all panels to homogenize the panels' axis and avoid misinterpretation.

**Additional remark independent from the referee comments:**

- We introduced a new title in the revised manuscript, "Mapping the safe operating space of marine ecosystems under contrasting emission pathways" and we added a brief scientific background related to the concept of safe operating space in the introduction section.
- We requested the addition of another co-author, Thorsten Blenckner, from the Stockholm Resilience Centre (Stockholm University, Sweden). Supported by the same EU COMFORT project, he has significantly contributed to the preparation of these replies to referees' comments, and he contributed substantially to revise the manuscript. We received the approval from the editorial board regarding to this request.
- We suggest not using the model CNRM-ESM2-1 in the plankton biomass metric due to a large inconsistent variability in this metric over the historical period (e.g., see figure below). After investigating the available outputs from this model in the ESGF portal, we found that the origin of this large variability is coming from the simulated mesozooplankton pool. We did not further investigate this issue.

[Figure]

Figure: Change in plankton biomass from the CMIP6 ensemble under the scenario SSP5-8.5 and relative to the 1850-1900 historical period as defined in the manuscript. The model ACCESS-ESM1-5 does not provide the necessary data to compute this metric.

**References**

Bindoff, N. L., Cheung, W. W. L., Kairo, J. G., Arístegui, J., Guinder, V. A., Hallberg, R., Hilmi, N. J. M., Jiao, N., Karim, M. S., Levin, L., O'Donoghue, S., Cuicapusa, S. R. P., Rinkevich, B., Suga, T., Tagliabue, A., and Williamson, P.: Changing Ocean, Marine Ecosystems, and Dependent Communities, in: IPCC Special Report on the Ocean and Cryosphere in a Changing Climate, edited by: Pörtner, H.-O., Roberts, D. C., Masson-Delmotte, V., Zhai, P., Tignor, M., Poloczanska, E., Mintenbeck, K., Alegría, A., Nicolai, M., Okem, A., Petzold, J., Rama, B., and Weyer, N. M., Cambridge University Press, Cambridge, UK and New York, NY, USA, 447–587, 2019.

Burger, F. A., Terhaar, J., and Frölicher, T. L.: Compound marine heatwaves and ocean acidity extremes, Nat Commun, 13, 4722, https://doi.org/10.1038/s41467-022-32120-7, 2022.

Capotondi, A., Rodrigues, R. R., Sen Gupta, A., Benthuysen, J. A., Deser, C., Frölicher, T. L., Lovenduski, N. S., Amaya, D. J., Le Grix, N., Xu, T., Hermes, J., Holbrook, N. J., Martinez-Villalobos, C., Masina, S., Roxy, M. K., Schaeffer, A., Schlegel, R. W., Smith, K. E., and Wang, C.: A global overview of marine heatwaves in a changing climate, Commun Earth Environ, 5, 701, https://doi.org/10.1038/s43247-024-01806-9, 2024.

Cocco, V., Joos, F., Steinacher, M., Frölicher, T. L., Bopp, L., Dunne, J., Gehlen, M., Heinze, C., Orr, J., Oschlies, A., Schneider, B., Segschneider, J., and Tjiputra, J.: Oxygen and indicators of stress for marine life in multi-model global warming projections, Biogeosciences, 10, 1849–1868, https://doi.org/10.5194/bg-10-1849-2013, 2013.

Deser, C., Phillips, A. S., Alexander, Michael. A., Amaya, D. J., Capotondi, A., Jacox, M. G., and Scott, J. D.: Future Changes in the Intensity and Duration of Marine Heat and Cold Waves: Insights from Coupled Model Initial-Condition Large Ensembles, Journal of Climate, 37, 1877–1902, https://doi.org/10.1175/JCLI-D-23-0278.1, 2024.

Deutsch, C., Penn, J. L., and Seibel, B.: Metabolic trait diversity shapes marine biogeography, Nature, 585, 557–562, https://doi.org/10.1038/s41586-020-2721-y, 2020.

Enting, I. G.: On the use of smoothing splines to filter $CO_2$ data, J. Geophys. Res., 92, 10977–10984, https://doi.org/10.1029/JD092iD09p10977, 1987.

Fröb, F., Bourgeois, T., Goris, N., Schwinger, J., and Heinze, C.: Simulated Abrupt Shifts in Aerobic Habitats of Marine Species in the Past, Present, and Future, Earth's Future, 12, e2023EF004141, https://doi.org/10.1029/2023EF004141, 2024.

Frölicher, T. L. and Laufkötter, C.: Emerging risks from marine heat waves, Nat Commun, 9, 650, https://doi.org/10.1038/s41467-018-03163-6, 2018.

Frölicher, T. L., Rodgers, K. B., Stock, C. A., and Cheung, W. W. L.: Sources of uncertainties in 21st century projections of potential ocean ecosystem stressors, Global Biogeochemical Cycles, 30, 1224–1243, https://doi.org/10.1002/2015GB005338, 2016.

Frölicher, T. L., Fischer, E. M., and Gruber, N.: Marine heatwaves under global warming, Nature, 560, 360–364, https://doi.org/10.1038/s41586-018-0383-9, 2018.

Hague, B. S., McGregor, S., Jones, D. A., Reef, R., Jakob, D., and Murphy, B. F.: The Global Drivers of Chronic Coastal Flood Hazards Under Sea-Level Rise, Earth's Future, 11, e2023EF003784, https://doi.org/10.1029/2023EF003784, 2023.

Jackson, L. C., Biastoch, A., Buckley, M. W., Desbruyères, D. G., Frajka-Williams, E., Moat, B., and Robson, J.: The evolution of the North Atlantic Meridional Overturning Circulation since 1980, Nat Rev Earth Environ, 3, 241–254, https://doi.org/10.1038/s43017-022-00263-2, 2022.

Jeltsch-Thömmes, A., Tran, G., Lienert, S., Keller, D. P., Oschlies, A., and Joos, F.: Earth system responses to carbon dioxide removal as exemplified by ocean alkalinity enhancement: tradeoffs and lags, Environ. Res. Lett., 19, 054054, https://doi.org/10.1088/1748-9326/ad4401, 2024.

Latif, M., Sun, J., Visbeck, M., and Hadi Bordbar, M.: Natural variability has dominated Atlantic Meridional Overturning Circulation since 1900, Nat. Clim. Chang., 12, 455–460, https://doi.org/10.1038/s41558-022-01342-4, 2022.

Lobelle, D., Beaulieu, C., Livina, V., Sévellec, F., and Frajka-Williams, E.: Detectability of an AMOC Decline in Current and Projected Climate Changes, Geophysical Research Letters, 47, e2020GL089974, https://doi.org/10.1029/2020GL089974, 2020.

Notz, D. and SIMIP community: Arctic Sea Ice in CMIP6, Geophysical Research Letters, 47, e2019GL086749, https://doi.org/10.1029/2019GL086749, 2020.

Oliver, E. C. J., Burrows, M. T., Donat, M. G., Sen Gupta, A., Alexander, L. V., Perkins-Kirkpatrick, S. E., Benthuysen, J. A., Hobday, A. J., Holbrook, N. J., Moore, P. J., Thomsen, M. S., Wernberg, T., and Smale, D. A.: Projected Marine Heatwaves in the 21st Century and the Potential for Ecological Impact, Front. Mar. Sci., 6, 734, https://doi.org/10.3389/fmars.2019.00734, 2019.

Oppenheimer, M., Glavovic, B. C., Hinkel, J., Wal, R., Magnan, A. K., Abd-ElGawad, A. M., Cai, R., Cifuentes-Jara, M., DeConto, R. M., Ghosh, T., Hay, J. E., Isla, F. I., Marzeion, B., Meyssignac, B., and Sebesvari, Z.: Sea Level Rise and Implications for Low-Lying Islands, Coasts and Communities, in: IPCC Special Report on the Ocean and Cryosphere in a Changing Climate, edited by: Pörtner, H. O., Roberts, D. C., Masson-Delmotte, V., Zhai, P., Tignor, M., Poloczanska, E., Mintenbeck, K., Alegría, A., Nicolai, M., Okem, A., Petzold, J., Rama, B., and Weyer, N. M., Cambridge University Press, Cambridge, UK and New York, NY, USA, 321–445, 2019.

Plecha, S. M. and Soares, P. M. M.: Global marine heatwave events using the new CMIP6 multi-model ensemble: from shortcomings in present climate to future projections, Environ. Res. Lett., 15, 124058, https://doi.org/10.1088/1748-9326/abc847, 2020.

Séférian, R., Berthet, S., Yool, A., Palmiéri, J., Bopp, L., Tagliabue, A., Kwiatkowski, L., Aumont, O., Christian, J., Dunne, J., Gehlen, M., Ilyina, T., John, J. G., Li, H., Long, M. C., Luo, J. Y., Nakano, H., Romanou, A., Schwinger, J., Stock, C., Santana-Falcón, Y., Takano, Y., Tjiputra, J., Tsujino, H., Watanabe, M., Wu, T., Wu, F., and Yamamoto, A.: Tracking Improvement in Simulated Marine Biogeochemistry Between CMIP5 and CMIP6, Curr. Clim. Change Rep., 6, 95–119, https://doi.org/10.1007/s40641-020-00160-0, 2020.

Smith, K. E., Burrows, M. T., Hobday, A. J., Sen Gupta, A., Moore, P. J., Thomsen, M., Wernberg, T., and Smale, D. A.: Socioeconomic impacts of marine heatwaves: Global issues and opportunities, Science, 374, eabj3593, https://doi.org/10.1126/science.abj3593, 2021.

Smith, K. E., Sen Gupta, A., Amaya, D., Benthuysen, J. A., Burrows, M. T., Capotondi, A., Filbee-Dexter, K., Frölicher, T. L., Hobday, A. J., Holbrook, N. J., Malan, N., Moore, P. J., Oliver, E. C. J., Richaud, B., Salcedo-Castro, J., Smale, D. A., Thomsen, M., and Wernberg, T.: Baseline matters: Challenges and implications of different marine heatwave baselines, Progress in Oceanography, 231, 103404, https://doi.org/10.1016/j.pocean.2024.103404, 2025.

Steinacher, M., Joos, F., and Stocker, T. F.: Allowable carbon emissions lowered by multiple climate targets, Nature, 499, 197–201, https://doi.org/10.1038/nature12269, 2013.

Terhaar, J., Kwiatkowski, L., and Bopp, L.: Emergent constraint on Arctic Ocean acidification in the twenty-first century, Nature, 582, 379–383, https://doi.org/10.1038/s41586-020-2360-3, 2020.

Terhaar, J., Vogt, L., and Foukal, N. P.: Atlantic overturning inferred from air-sea heat fluxes indicates no decline since the 1960s, Nat Commun, 16, 222, https://doi.org/10.1038/s41467-024-55297-5, 2025.

Tittensor, D. P., Novaglio, C., Harrison, C. S., Heneghan, R. F., Barrier, N., Bianchi, D., Bopp, L., Bryndum-Buchholz, A., Britten, G. L., Büchner, M., Cheung, W. W. L., Christensen, V., Coll, M.,

Dunne, J. P., Eddy, T. D., Everett, J. D., Fernandes-Salvador, J. A., Fulton, E. A., Galbraith, E. D., Gascuel, D., Guiet, J., John, J. G., Link, J. S., Lotze, H. K., Maury, O., Ortega-Cisneros, K., Palacios-Abrantes, J., Petrik, C. M., du Pontavice, H., Rault, J., Richardson, A. J., Shannon, L., Shin, Y.-J., Steenbeek, J., Stock, C. A., and Blanchard, J. L.: Next-generation ensemble projections reveal higher climate risks for marine ecosystems, Nat. Clim. Chang., 11, 973–981, https://doi.org/10.1038/s41558-021-01173-9, 2021.

Weaver, A. J., Sedláček, J., Eby, M., Alexander, K., Crespin, E., Fichefet, T., Philippon-Berthier, G., Joos, F., Kawamiya, M., Matsumoto, K., Steinacher, M., Tachiiri, K., Tokos, K., Yoshimori, M., and Zickfeld, K.: Stability of the Atlantic meridional overturning circulation: A model intercomparison, Geophys. Res. Lett., 39, 2012GL053763, https://doi.org/10.1029/2012GL053763, 2012.

Weijer, W., Cheng, W., Garuba, O. A., Hu, A., and Nadiga, B. T.: CMIP6 Models Predict Significant 21st Century Decline of the Atlantic Meridional Overturning Circulation, Geophys. Res. Lett., 47, e2019GL086075, https://doi.org/10.1029/2019GL086075, 2020.

---

## Author Response (AR2)

**Author's response to referee comment on egusphere-2024-2768**

We thank Anonymous referee #2 for this second round of thoughtful comments and suggestions. We have addressed all points raised in the review as detailed in our point-by-point response below. Please note additional remarks independent from the referee comments, placed at the end of this document.

The referee comment is shown in italic grey font while our reply including the relevant changes made in the manuscript are shown in black font. Line numbers refer to the location in the version of the manuscript that the referee reviewed (i.e., before the hereby edits, version without track-changes).

**Report #1, Submitted on 28 Jul 2025, Anonymous referee #2**

*I have previously reviewed Bourgeois et al., which conducted an extensive literature review to define 15 marine ecosystem impact metrics and 4 levels of thresholds (which they call "mitigation limits") corresponding to each metric, and then used CMIP6 ESMs and two additional ESMICs to evaluate the probability of each metric staying/exceeding each threshold under different mitigation scenario. I am overall satisfied with and appreciate the amount of efforts that the authors made to address my earlier comments, and see much improvement in the quality of the revised manuscript.*

Thank you for this positive feedback.

*I have one more suggestion to make, though, to further improve the clarify of the abstract/intro/methods of the manuscript, as I find these sections are quite difficult for the readers to follow unless the readers jump to the Results (e.g., Table 1) first. Examples are Lines 26-29, Line 67, Lines 85-88, sentences hard to understand even for myself as a second-time reader of this manuscript.*

We rephrased and split sentences at the lines mentioned to improve the readability (see further down for detailed changes).

*As mentioned above, the manuscript defines: 15 marine ecosystem relevant impact metrics 4 levels of thresholds exceeding which the marine ecosystem may be at risk (first of all, I think it is more appropriate to call these thresholds "mitigation targets" or just "thresholds" than "mitigation limits", the latter of which sounds more related to factors that limit a mitigation strategy reaches its targets.)*

The choice of using "threshold", "mitigation target", or "mitigation limit" has been a topic of discussion within our team in the early step of writing the manuscript. We initially used the term "mitigation target".
Without context, a target is a value that is desirable to reach while a limit is a value to avoid or stay within. The latter is more in line with what we aim to express. The term "target" is rather used for commitments related to climate and energy policy, such as nationally

determined contributions, and carbon neutrality under the UNFCCC[1] framework. As for the term "threshold", it means semantically that exceeding a certain limit would trigger an event (e.g., abrupt change, tipping point). This is not necessarily the case in our study where our limits are express a gradual increase in severity of impacts induced by a wide range of long-term ocean changes due to climate change, including (but not limited to) tipping points and abrupt changes. However, we see the referee's point that mitigation limit "sounds more related to factors that limit a mitigation strategy reaches its targets". Thus, we suggest keeping the use of the term "limit" but agree on removing the word "mitigation". Considering the multiple occurrence of the expression "mitigation limit", we do not list below the locations where the word "mitigation" has been removed and we refer the reviewer to the track-changes version of the manuscript.

*The manuscript also evaluates model simulations under 3 emission scenarios: SSP1-2.6, SSP-3.4 and SSP5-8.5. Finally, the manuscript computes the probability (>80% vs. 50-80%) of each metric exceeding a certain threshold. There are together 15\*4\*3 = 180 probability (and when that will happen) to compute, and each can be phrased as a science question, such as: "Under the highest emission scenario considered, by year 2100, how likely is it for global mean sea level to rise 0.4 m compared to 1850-1900, and when will that happen?". I suggest that authors throw out questions like this as early as possible in the manuscript (no later than Intro), then state that in order to answer this type of questions we need to define XX, YY, ZZ, and need data to compute xx, yy, zz.*

We agree that stating explicitly scientific questions in the introduction section improves the clarity of the manuscript. We followed the referee's suggestion (see further down for detailed changes, edits on lines 66–72).

*Also clearly define that the targets level 1-4 are from more ambitious (challenging to reach) to more relaxed.*

This aspect is already defined in lines 84-87 (second sentence of the methods section), but we have now replicated the referee's wording at the end the introduction (see further down for detailed changes, edits on lines 66–72).

*Break long sentences into shorter ones where it is possible. I hope (and believe) this will help improve the readability of the manuscript.*

As mentioned above, we rephrased and split sentences not only at the lines mentioned earlier but also in other instances where appropriate to improve the readability (see further down for detailed changes).
* * *
[1] United Nations Framework Convention on Climate Change

Lines 26–28 (rewording)

*Before*

[revised manuscript text omitted]

Additional typographic and small edits independent from the referee comments

- Affiliation for NORCE researchers updated following a change of legal name (NORCE Research)
- Use of dashes for ranges of values, instead of hyphens (lines 114, 118, 125, 137–138, 141, and 152–153, 198, 221, 222, 266, 349, 373, 414–415, 496 and in Table 1)
- Medium and high probability were sometimes inaccurately referred in the text as "50–80 %" and ">80 %", respectively. "50–79 %" and "≥ 80 %" are now used consistently when referring to medium and high probability (edits in lines 349, 358, 369, 391, 414, 496). These edits do not change the results nor the figures.
- Addition of the sentences below to enrich the interpretation of ESM-EMIC comparison at line 441:
  "Another explanation could be how EMIC ensembles are constrained. For UVic, global mean profiles of ocean tracers have been used as observational constraints. The latter averages large regional variations that compensate each other resulting in similar global mean. Such globally averaged constraints might inefficiently reduce uncertainty for ice-dominated polar regions, especially since we did not constrain sea-ice area. Large variations in sea-ice cover could influence air-sea gas exchange and, as a result, Arctic Ocean $\Omega_a < 1$."
- Add missing spaces before "%" or after "≥" symbols (lines 161, 349, 350, 354-360, 371, 391, 414, 476, 496, 508, 589).
- Use of consecutive figure numbering in the appendix from "Figure A1" to "Figure A7" as required by the editorial board.
- Acknowledgments of the reviewers and the associate editor are added.